# β-Catenin-NF-κB-CFTR interactions in cholangiocytes regulate inflammation and fibrosis during ductular reaction

Shikai Hu[1,2†], Jacquelyn O Russell[2†], Silvia Liu[2,3], Catherine Cao[2], Jackson McGaughey[2], Ravi Rai[2], Karis Kosar[2], Junyan Tao[2], Edward Hurley[4], Minakshi Poddar[2], Sucha Singh[2], Aaron Bell[2], Donghun Shin[3,5], Reben Raeman[2,3], Aatur D Singhi[2,3], Kari Nejak-Bowen[2,3], Sungjin Ko[2,3*], Satdarshan P Monga[2,3,6*]

[1]School of Medicine, Tsinghua University, Beijing, China; [2]Department of Pathology, University of Pittsburgh School of Medicine and University of Pittsburgh Medical Center, Pittsburgh, United States; [3]Pittsburgh Liver Research Center, University of Pittsburgh School of Medicine and University of Pittsburgh Medical Center, Pittsburgh, United States; [4]Department of Pediatrics, University of Pittsburgh School of Medicine and University of Pittsburgh Medical Center, Pittsburgh, United States; [5]Department of Developmental Biology, University of Pittsburgh School of Medicine and University of Pittsburgh Medical Center, Pittsburgh, United States; [6]Department of Medicine, University of Pittsburgh School of Medicine and University of Pittsburgh Medical Center, Pittsburgh, United States

*For correspondence:
sungjin@pitt.edu (SK);
smonga@pitt.edu (SPM)

[†]These authors contributed equally to this work

Competing interest: The authors declare that no competing interests exist.

**Abstract** Expansion of biliary epithelial cells (BECs) during ductular reaction (DR) is observed in liver diseases including cystic fibrosis (CF), and associated with inflammation and fibrosis, *albeit* without complete understanding of underlying mechanism. Using two different genetic mouse knockouts of β-catenin, one with β-catenin loss is hepatocytes and BECs (KO1), and another with loss in only hepatocytes (KO2), we demonstrate disparate long-term repair after an initial injury by 2-week choline-deficient ethionine-supplemented diet. KO2 show gradual liver repopulation with BEC-derived β-catenin-positive hepatocytes and resolution of injury. KO1 showed persistent loss of β-catenin, NF-κB activation in BECs, progressive DR and fibrosis, reminiscent of CF histology. We identify interactions of β-catenin, NFκB, and CF transmembranous conductance regulator (CFTR) in BECs. Loss of CFTR or β-catenin led to NF-κB activation, DR, and inflammation. Thus, we report a novel β-catenin-NFκB-CFTR interactome in BECs, and its disruption may contribute to hepatic pathology of CF.

## Introduction

The liver possesses unique regenerative potential. During chronic liver injury, however, liver fibrosis accompanies regeneration and can progress to cirrhosis, which can then progress to end-stage liver disease (ESLD) or hepatocellular cancer (HCC) (*Pellicoro et al., 2014*). Currently, cirrhosis is the 11th leading cause of death globally, and the incidence of liver disease continues to rise as conditions such as non-alcoholic fatty liver disease (NAFLD) and alcoholic liver disease continue to prevail (*Asrani et al., 2019*). Thus, there has been great interest in studying mechanisms of injury, inflammation, and fibrosis during liver injury in order to effectively develop novel therapies. The role of hepatic epithelial cells (referred henceforth as 'hepithelial' cells), which include both hepatocytes and cholangiocytes or biliary epithelial cells (BECs), in regulating microenvironment is beginning to be appreciated. Loss of hepatocyte differentiation in chronic liver diseases and ESLD, either due to much needed hepithelial

**eLife digest** The liver has an incredible capacity to repair itself or 'regenerate' – that is, it has the ability to replace damaged tissue with new tissue. In order to do this, the organ relies on hepatocytes (the cells that form the liver) and bile duct cells (the cells that form the biliary ducts) dividing and transforming into each other to repair and replace damaged tissue, in case the insult is dire.

During long-lasting or chronic liver injury, bile duct cells undergo a process called 'ductular reaction', which causes the cells to multiply and produce proteins that stimulate inflammation, and can lead to liver scarring (fibrosis). Ductular reaction is a hallmark of severe liver disease, and different diseases exhibit ductular reactions with distinct features. For example, in cystic fibrosis, a unique type of ductular reaction occurs at late stages, accompanied by both inflammation and fibrosis. Despite the role that ductular reaction plays in liver disease, it is not well understood how it works at the molecular level.

Hu et al. set out to investigate how a protein called β-catenin – which can cause many types of cells to proliferate – is involved in ductular reaction. They used three types of mice for their experiments: wild-type mice, which were not genetically modified; and two strains of genetically modified mice. One of these mutant mice did not produce β-catenin in biliary duct cells, while the other lacked β-catenin both in biliary duct cells and in hepatocytes.

After a short liver injury – which Hu et al. caused by feeding the mice a specific diet – the wild-type mice were able to regenerate and repair the liver without exhibiting any ductular reaction. The mutant mice that lacked β-catenin in hepatocytes showed a temporary ductular reaction, and ultimately repaired their livers by turning bile duct cells into hepatocytes. On the other hand, the mutant mice lacking β-catenin in both hepatocytes and bile duct cells displayed sustained ductular reactions, inflammation and fibrosis, which looked like that seen in patients with liver disease associated to cystic fibrosis. Further probing showed that β-catenin interacts with a protein called CTFR, which is involved in cystic fibrosis. When bile duct cells lack either of these proteins, another protein called NF-B gets activated, which causes the ductular reaction, leading to inflammation and fibrosis.

The findings of Hu et al. shed light on the role of β-catenin in ductular reaction. Further, the results show a previously unknown interaction between β-catenin, CTFR and NF-B, which could lead to better treatments for cystic fibrosis in the future.

---

proliferation for repair, or as an adaptation to escape injury, seems to contribute to not only loss of key hepatic functions, but is also causally associated with increased immune response and hepatic fibrosis (*Argemi, 2019*; *Nishikawa, 2015*). However, how hepithelial cells may modulate hepatic immune microenvironment is unclear.

As an important hepithelial cell type, BECs are known to undergo proliferation to replace dying BECs in cholangiopathies or cystic liver diseases, as well as can under phenotypic switch to generate de novo hepatocytes when hepatocytes are chronically injured and/or are unable to optimally proliferate, phenomena termed as ductular reaction (DR) (*Sato et al., 2019*; *Wilson and Rudnick, 2019*). Reactive ductules, however, can secrete pro-inflammatory and pro-fibrotic cytokines to induce inflammation, activate myofibroblasts, and induce fibrosis. The extent of DR correlates with fibrosis in many types of liver injuries (*Lowes et al., 1999*; *Richardson, 2007*; *Zhao et al., 2018*). The molecular underpinnings of reactive DR are incompletely understood although molecules like Yes-associated protein-1 (YAP1) have been implicated (*Planas-Paz, 2019*).

β-Catenin, the major downstream effector of the Wnt signaling, is a well-known mediator of hepatocyte proliferation. Liver-specific (hepatocyte and BECs) β-catenin knockout (KO1) mice generated by breeding β-catenin-floxed and albumin-cre mice show delayed liver regeneration (LR) after partial hepatectomy or after toxicant-induced liver injury (*Apte, 2009*; *Tan et al., 2006*). When KO1 were administered choline-deficient, ethionine-supplemented (CDE) diet, it triggered greater steatosis, cell death, DR, inflammation, and fibrosis than wild-type (WT1), and upon switching to normal diet for 2 weeks for recovery, continued to show greater injury due to an impairment of hepatocyte proliferation (*Akhurst, 2001*; *Russell, 2019*). Similar greater injury, fibrosis, and DR were observed in CDE-fed hepatocyte-only β-catenin KO (KO2), generated by delivering adeno-associated virus serotype 8 carrying a plasmid encoding *Cre* recombinase under a hepatocyte-specific thyroxine-binding globulin

(TBG) promoter (AAV8-TBG-Cre) into the β-catenin-floxed mice (*Russell, 2019*). Intriguingly, labeling BECs for fate-tracing showed the liver repair to occur through BEC-to-hepatocyte transdifferentiation upon recovery for 2 weeks and up to 6 months on normal diet, although long-term impact on injury resolution, inflammation, DR, and fibrosis was not studied in either model (*Russell, 2019*).

In the current study, we investigate hepatic injury and repair in KO2 and KO1 mice challenged for 2 weeks with CDE diet and allowed to recover on normal diet for 2 weeks, 3 months, and 6 months. Intriguingly, we observed highly divergent injury-repair responses in the two models. KO2 mice showed progressive repair through expansion of BEC-derived β-catenin-positive hepatocytes and resolution of inflammation, DR, and fibrosis. However, KO1 display progressive and peculiar DR composed of numerous small luminal structures lined by a single layer of BECs, even after being on normal diet for 6 months, which is associated with fibrosis and inflammation, and is reminiscent of cystic fibrosis (CF)-like morphology. We identify a unique interactome of β-catenin, p65 subunit of NF-κB and cystic fibrosis transmembranous conductance regulator (CFTR) in BECs and show perturbations in these interactions, leading to excessive NF-κB activation and inflammation in BECs in both KO1 and CF patients.

## Results

### Long-term follow-up of mice lacking β-catenin in hepatocytes only (KO2) shows delayed but eventual resolution of fibrosis and DR after initial 2-week CDE diet

We previously showed CDE diet for 2 weeks led to enhanced injury, fibrosis, and DR in mice lacking β-catenin in hepatocytes only (KO2), generated by delivering AAV8-TBG-Cre into *Ctnnb1*$^{flox/flox}$; *Rosa-stop*$^{flox/flox}$-*EYFP* mice, as compared to WT2 mice, generated by injecting AAV8-TBG-Cre into *Ctnnb1*$^{+/+}$; *Rosa-stop*$^{flox/flox}$-*EYFP* mice (*Russell, 2019*). And that upon switching to normal diet, liver repair occurred via hepatocyte proliferation in WT2 but through BEC-to-hepatocyte transdifferentiation in KO2 (*Russell, 2019*). To specifically investigate durability of repair especially after the increased injury observed in the KO2 mice at 2 weeks of CDE diet, we fed CDE diet to KO2 and WT2 mice for 2 weeks and switched to normal diet for 2 weeks, 3 months, or 6 months (*Figure 1A*). KO2 mice had elevated serum alanine aminotransferase (ALT) and total bilirubin (BR) levels than WT2 at 2 weeks of CDE diet, but returned to normal at 2 weeks onwards after switching to normal diet, similar to WT2, although BR levels tended to be higher in KO2 up to 3 months of recovery (*Figure 1B*). Alkaline phosphatase (ALP) was increased in WT2 and KO2 after 2 weeks of CDE injury but returned to normal at 2 weeks of recovery in both groups (*Figure 1B*).

Previously by fate-tracing, we observed a BEC transdifferentiated to hepatocytes and expanded in KO2 (*Russell, 2019*). Likewise, by immunohistochemistry (IHC) for β-catenin that is only present in BECs in KO2 at baseline, there were increased numbers of β-catenin-positive hepatocytes at 3 months and 6 months of recovery (*Figure 1C*, *Figure 1—figure supplement 1*). RT-PCR for β-catenin gene (*Ctnnb1*) expression showed increasing expression in KO2 livers over time, becoming comparable to WT2 at 6 months (*Figure 1D*). Since glutamine synthetase (GS) is a β-catenin-specific target, we also examined its protein by IHC in WT2 and KO2 at the 6- month recovery time. Indeed, like in WT2, zone 3 GS-positive cells are present in KO2 showing β-catenin presence and activation at this time point (*Figure 1—figure supplement 2*).

Sirius Red staining for fibrosis showed greater collagen deposition in KO2 than WT2 at 2 weeks of CDE diet and persisted at 2 weeks of recovery (*Figure 2A and B*). Interestingly, despite being on normal diet and lack of any ongoing injury, KO2 continued to show fibrosis at 3 months, eventually resolving at 6 months (*Figure 2A and B*). Likewise, expression of *Col1a1* and *Tgfβ2* tended to be higher in KO2 compared to WT2 mice at 3- month recovery, but were comparable to WT2 at 6 months (*Figure 2C*).

Since increased DR was observed in KO2 after CDE diet-induced injury, and fibrosis can be associated with DR, we next performed IHC for pan-cytokeratin (PanCK; *Figure 2D*). There was robust DR in KO2 mice and WT2 mice after 2 -week CDE diet and was also evident at 2 weeks of recovery although it appeared to be more pronounced in KO2. At 3 months of recovery, normal bile ducts are seen in WT2, whereas DR composed of flattened, non-luminal, and single or few cell clusters is evident throughout liver lobule in KO2 (*Figure 2D*). At 6 months, there was no DR in either group (*Figure 2D*).

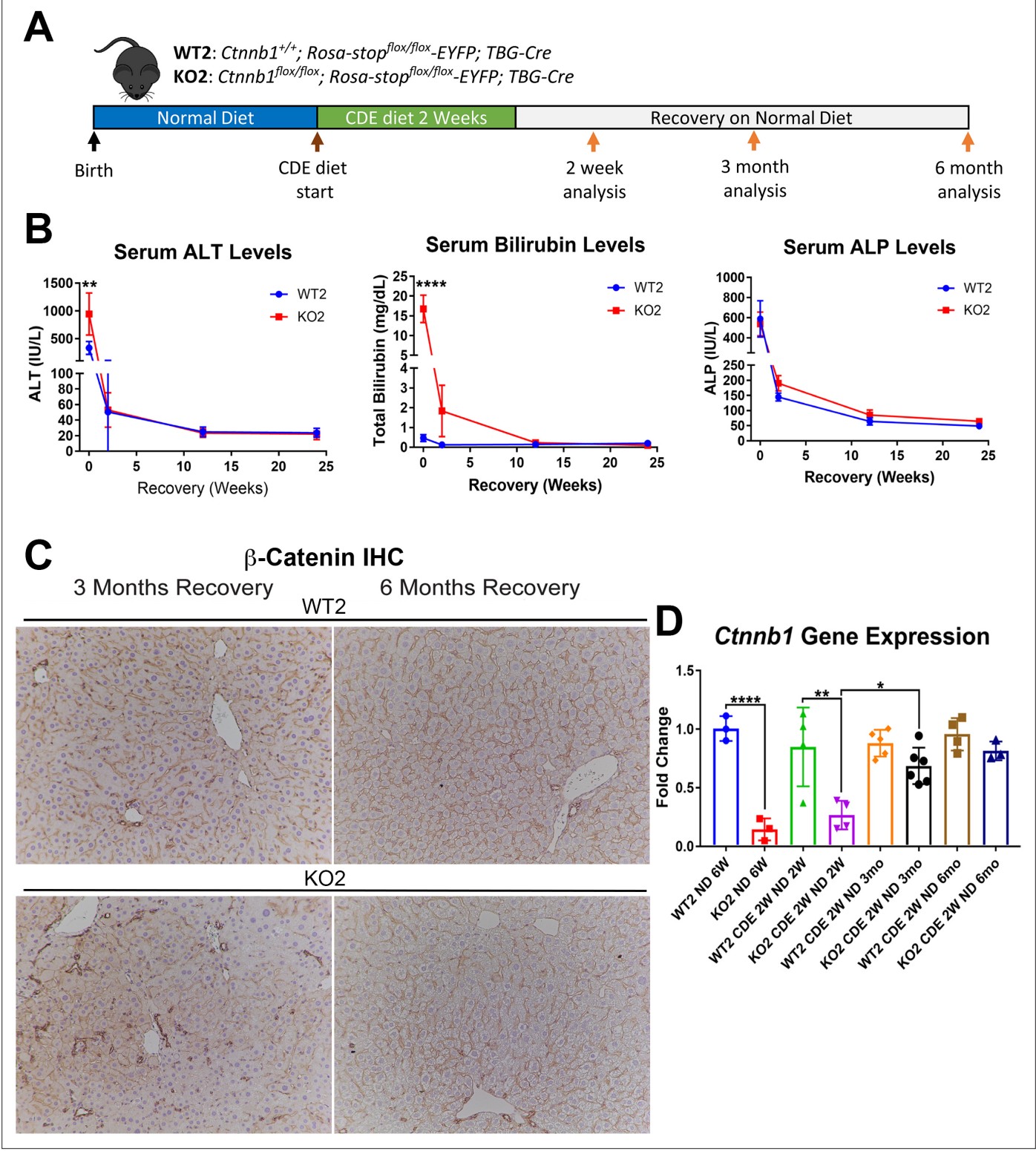

**Figure 1.** Comparable recovery of WT2 and KO2 on normal diet after initial 2-week choline-deficient, ethionine-supplemented (CDE) diet injury, along with repopulation of KO2 livers with biliary epithelial cell (BEC)-derived β-catenin-positive hepatocytes. (**A**) Experimental design showing WT2 and KO2 on 2 weeks of CDE diet and recovery on normal diet for up to 6 months with analysis at intermediate time points as indicated. (**B**) Serum alanine aminotransferase (ALT), bilirubin, and alkaline phosphatase (ALP) in the two groups over time (one-way ANOVA, **p<0.01, ****p<0.0001, n = 3–6 per group). (**C**) β-Catenin immunohistochemistry in WT2 and KO2 mice at 3 months and 6 months of recovery showing β-catenin-positive BECs and hepatocytes in KO2 and WT2 (100×). (**D**) *Ctnnb1* gene expression in WT2 and KO2 mice during recovery from CDE diet (one-way ANOVA, *p<0.05,

*Figure 1 continued on next page*

*Figure 1 continued*

**p<0.01, ****p<0.0001. n = 3–6 per group; individual animal values represented by dots).

The online version of this article includes the following figure supplement(s) for figure 1:

**Figure supplement 1.** β-Catenin immunohistochemistry in WT2 and KO2 mice at 3 months and 6 months of recovery showing β-catenin-positive biliary epithelial cells (BECs) and hepatocytes in KO2 and WT2 (200× ).

**Figure supplement 2.** Immunohistochemistry for glutamine synthetase (GS) in WT2 and KO2 mice at 6 months of recovery showing KO2 beginning to express GS in the zone-3 hepatocytes as biliary epithelial cell (BEC)-derived β-catenin-positive hepatocyte repopulate these livers at 6 months after recovery from 2- week choline-deficient, ethionine-supplemented (CDE) injury (100× ).

Quantification of PanCK staining was done using NIH Imager as described in Methods and showed higher trends but insignificant differences in areas covered by PanCK-positive DR in KO2 than WT2 at all times (*Figure 2—figure supplement 1*). Gene expression of BEC markers *Krt19* and *Epcam* supported these observations (*Figure 2E*).

Expression of the gene encoding tissue inhibitor of metalloproteinase 1a (*Timp1*), a well-known inhibitor of matrix metalloproteinases, known for a role in extracellular matrix degradation, was determined next as a possible mechanism of fibrosis resolution (*Yoshiji, 2000*; *Yoshiji, 2002*). Higher expression of *Timp1* persisted in KO2 as compared to WT2 at all recovery times except 6 months, coinciding with resolution of fibrosis and DR (*Figure 2F*).

Taken together, these results suggest that higher DR is associated with greater fibrosis in KO2, and resolution of the DR and fibrosis took longer in KO2 than WT2, which correlated with enhanced repopulation of the KO2 liver with β-catenin-positive hepatocytes and normalization of Timp1 levels.

## Long-term follow-up of mice lacking β-catenin in hepatocytes and BECs (KO1) shows prolonged fibrosis and DR without any evidence of regression after initial 2- week CDE diet

Next, we placed *Albumin-Cre^+/- Ctnnb1^flox/flox* (KO1) mice lacking β-catenin in hepatocytes and BECs, and their wild-type littermates (WT1) on CDE diet for 2 weeks and allowed recovery on normal diet for 2 weeks, 3 months, and 6 months (*Figure 3A*). We observed severe liver injury in KO1 mice, shown by significantly higher serum ALT and total BR after 2 weeks of CDE diet compared to WT1. During recovery, serum ALT levels in KO1 and WT1 mice decreased to normal levels (*Figure 3B*), while BR remained mildly elevated in KO1 mice up to 3 months of recovery as compared to WT1, which returned to normal at 2 weeks of recovery (*Figure 3B*). Serum ALP levels were comparably increased in WT1 and KO1 at 2 weeks of CDE injury and returned to normal levels at 2 -week recovery (*Figure 3B*).

Since β-catenin is lacking in hepithelial cells in the KO1 livers, IHC for β-catenin and RT-PCR for *Ctnnb1* showed continued absence in KO1 and not WT1 at 3- month and 6- month recovery on normal diet (*Figure 3C and D*, *Figure 3—figure supplement 1*). As a surrogate for β-catenin presence and activity, we also assessed GS by IHC in WT1 and KO1 at the 6- month recovery time. Unlike WT1, there is a complete absence of GS in zone 3 hepatocytes in KO1 showing continued absence of β-catenin at this time point (*Figure 3—figure supplement 2*).

We previously reported increased fibrosis in KO1 mice compared to WT1 littermates after 2- week CDE diet (*Russell, 2019*). Here, we evaluated fibrosis during recovery on normal diet in both WT1 and KO1. Despite normalization of serum transaminases during recovery, we observed continued fibrosis especially in the periportal area in KO1 and especially at 3 months and 6 months by Sirius Red staining, whereas WT1 mice displayed resolution of fibrosis as early as 2- week recovery (*Figure 4A*). Quantification verified significant increases in fibrosis in KO1 at all time points compared to WT1 (*Figure 4B*). Additionally, expression of *Col1a1* tended to be higher in KO1 mice during recovery (*Figure 4C*).

DR was next assessed by IHC for PanCK. While there was a dramatic decrease in DR overtime in WT1 mice, a profound DR was observed in KO1 mice at all times, which was even more pronounced at 6 months of recovery (*Figure 4D*). Quantification of PanCK staining showed significant differences in areas covered by PanCK-positive DR in KO1 than WT1 at all times of recovery from CDE diet and even significant and progressive increase from 3 months to 6 months in KO1 (*Figure 4—figure supplement 1*). Furthermore, the DR was peculiar and composed of numerous small luminal structures lined by a single layer of PanCK-positive columnar cells at 3 months and 6 months, rather than more flattened

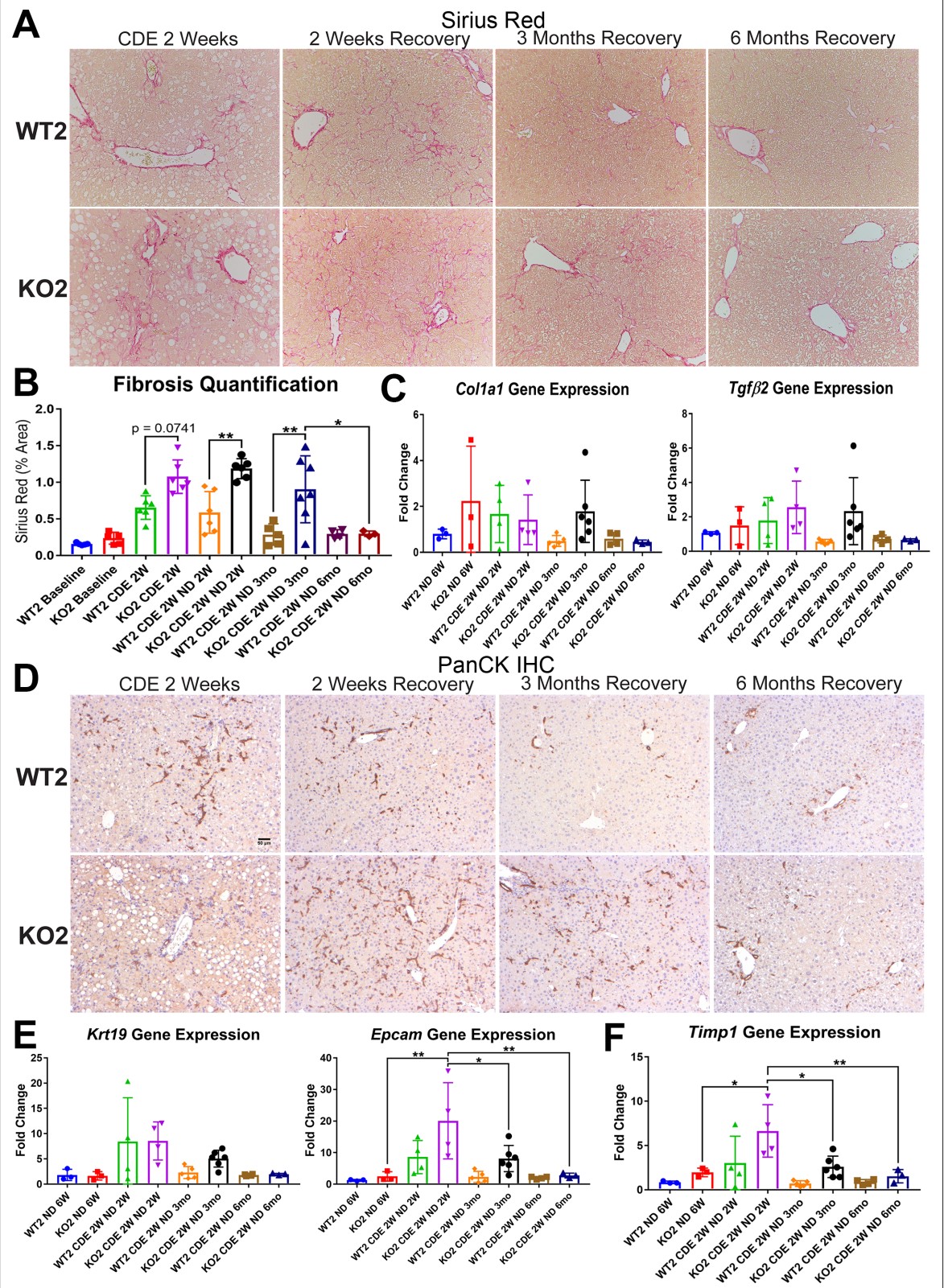

**Figure 2.** Fibrosis and ductular reaction (DR) is sustained in KO2 mice recovering on normal diet until 3 months, but subsides by 6 months, after initial 2- week choline-deficient, ethionine-supplemented (CDE) diet. (**A**) Sirius Red staining in WT2 and KO2 mice over time during recovery from CDE diet (100× ). (**B**) Quantification of Sirius Red staining (one-way ANOVA, *p<0.05, **p<0.01. n = 3–7 per group; individual animal values represented by dots). (**C**) A trend of increased expression of *Col1a1* and *Tgfβ2* in KO2 mice at 3 months of recovery but not at 6 months (n = 3–6 per group; individual animal

*Figure 2 continued on next page*

*Figure 2 continued*

values represented by dots). (**D**) Pan-cytokeratin (PanCK) staining in WT2 and KO2 mice over time during recovery from CDE diet. Scale bar = 50 µm. (**E**) A trend of higher *Krt19* expression and significantly higher expression of *Epcam* gene in KO2 up to 3 months on recovery and normalization to WT2 levels at 6 months (one-way ANOVA, *p<0.05, **p<0.01, n = 3–6 per group, individual animal values represented by dots). (**F**) Significantly higher *Timp1* gene expression in KO2 than WT2 up to 3 months on recovery diet and normalization at 6 months (one-way ANOVA, *p<0.05, **p<0.01, n = 3–6 per group, individual animal values represented by dots).

The online version of this article includes the following figure supplement(s) for figure 2:

**Figure supplement 1.** Similar ductular reaction (DR) during recovery from choline-deficient, ethionine-supplemented (CDE) in KO2 compared with WT2.

and invasive DR without lumen seen at earlier stages of CDE injury and recovery in both WT1, KO1, and even KO2 (*Figures 4D and 2D*). Enhanced gene expression for *Krt19* and *Epcam* was simultaneously evident in KO1 at these times (*Figure 4E*).

To determine if the continued DR was due to ongoing BEC proliferation, we co-stained KO1 livers from 3- month recovery with PanCK and proliferating cell nuclear antigen (PCNA) (*Figure 4— figure supplement 2A*). Significantly more BECs were proliferating in KO1 compared to WT1 mice (*Figure 4—figure supplement 2B*). A subset of BECs in DR were also positive for phospho-Erk1/2 (p-Erk1/2), known for regulating BEC proliferation (*Figure 4—figure supplement 2C*; *Pepe-Mooney, 2019*).

Since decreased gene expression of *Timp1* correlated with reduced fibrosis in KO2 at 6 months of recovery, we next investigated its levels in KO1 and WT1. Timp1 tended to be upregulated in KO1 mice at all times but significantly at 6- month recovery time (*Figure 4F*).

Taken together, these results suggest that KO1 mice that continue to lack β-catenin in hepithelial cells show persistent Timp1 and fibrosis, and display continued and morphologically distinct DR associated with increased BEC proliferation at all times after the initial 2- week CDE diet injury, despite lack of active insult.

## Unremarkable changes in hepatic bile acids, apoptosis, and senescence during resolution of fibrosis and DR during recovery from CDE diet in KO2 mice

To discern the basis of disparate DR and fibrosis between the two models, we first focused on investigating differences in specific injury processes between KO2 and WT2 during CDE injury and recovery. Increased hepatic bile acids have been implicated in hepatic injury and repair (*Fickert and Wagner, 2017*), and our lab has previously reported altered hepatic bile acids (BAs) in KO1 mice after methionine-choline-deficient (MCD) diet (*Behari, 2010*; *Thompson, 2018*). However, CDE diet-fed KO2 or WT2 mice showed no significant increase in hepatic bile acids at any time, suggesting these to not be driving DR or fibrosis in this model (*Figure 4—figure supplement 3A*). We also investigated cell senescence as a possible driver of DR and fibrosis (*Wei-Yu et al., 2015*). However, no significant hepatocyte senescence was observed by p21 IHC in WT2 or KO2 mice during long-term recovery from CDE diet (*Figure 4—figure supplement 3B*).

Although ALTs were not elevated, we wanted to directly address any ongoing injury in recovering WT1 and KO1 mice. Cleaved caspase 3 staining showed minimal cell death in both WT1 and KO1 mice at 3 months of recovery (*Figure 4—figure supplement 3C*).

Thus, BA alterations, cellular senescence, and cell death are not the basis of DR and fibrosis in CDE injury, and hence cannot explain differences in recovery between KO1 and KO2 mice.

## Maintenance of adherens junctions in KO2 and KO1 during recovery from CDE diet injury

Next, we assessed if continued absence of β-catenin in KO1 but not in KO2 at 6 months of recovery could be affecting adherens junctions (AJs) integrity and could explain differences in DR and fibrosis between KO2 and KO1 mice. Temporal disruption of cell-cell junctions has been associated with pathologies in hepatobiliary injury including after CDE diet (*Pradhan-Sundd, 2018a*). However, at 6 months of recovery, immunoprecipitation (IP) with E-cadherin showed an association of E-cadherin to β-catenin in KO2 and with γ-catenin in KO1, as has been shown in β-catenin-deficient livers by us

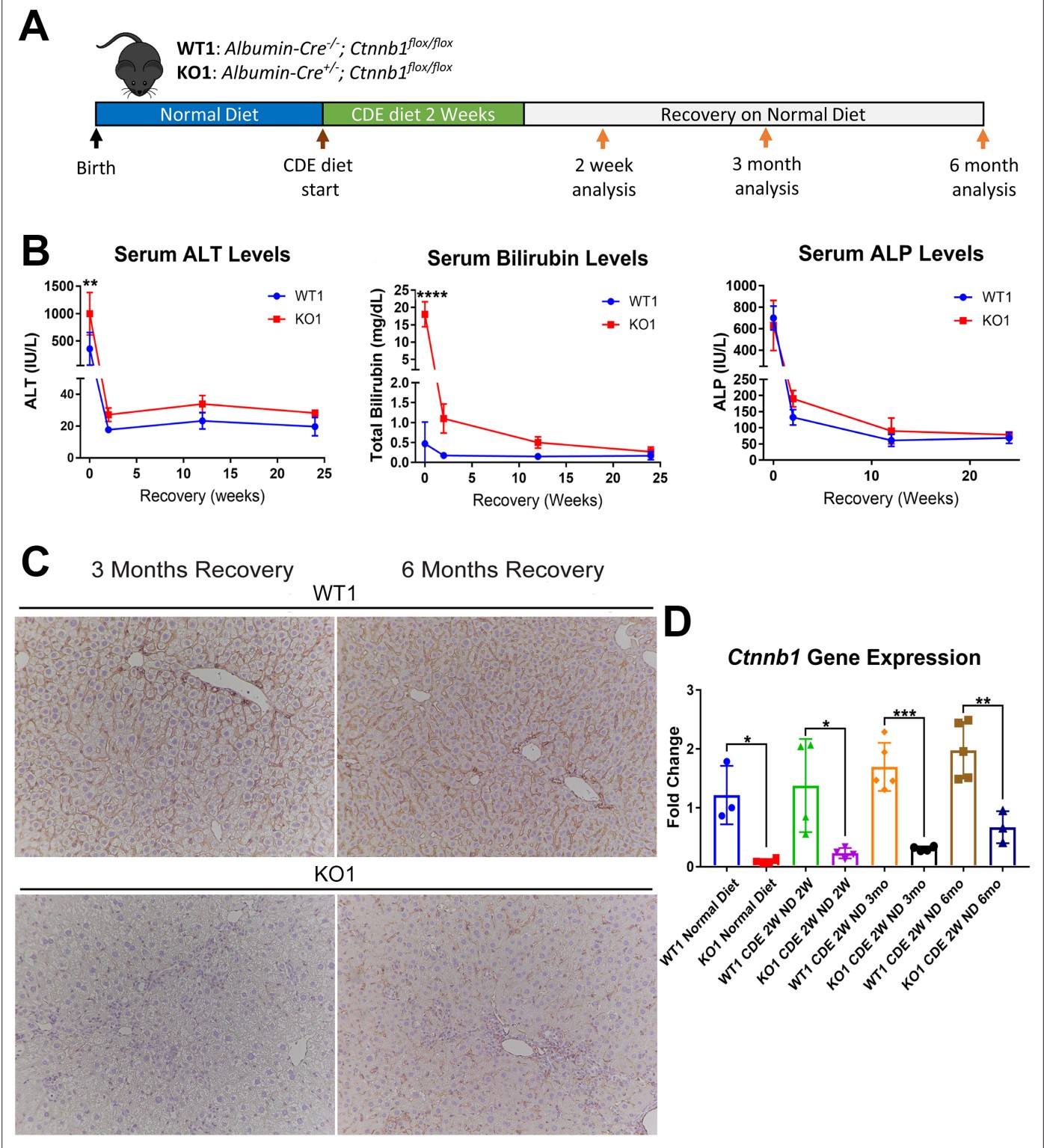

**Figure 3.** Serum biochemistry suggests comparable recovery on normal diet in WT1 and KO1 after 2- week choline-deficient, ethionine-supplemented (CDE) diet and continued lack of β-catenin in KO1. (**A**) Experimental design showing WT1 and KO1 on 2 weeks of CDE diet and recovery on normal diet for up to 6 months with analysis at intermediate time points as indicated. (**B**) Serum alanine aminotransferase (ALT), bilirubin, and alkaline phosphatase (ALP) in the two groups over time (one-way ANOVA, **p<0.01, ****p<0.0001, n = 3–5 per group). (**C**) β-Catenin immunohistochemistry in WT1 and KO1 mice at 3 months and 6 months of recovery showing absence of β-catenin in biliary epithelial cells (BECs) and hepatocytes in KO1 (100× ). (**D**) *Ctnnb1* gene expression in WT1 and KO1 mice during recovery from CDE diet shows continued β-catenin absence over time in KO1 (one-way ANOVA, *p<0.05,

*Figure 3 continued on next page*

*Figure 3 continued*

**p<0.01, ****p<0.0001, n = 3–5 per group, individual animal values represented by dots).

The online version of this article includes the following figure supplement(s) for figure 3:

**Figure supplement 1.** β-Catenin immunohistochemistry in WT1 and KO1 mice at 3 months and 6 months of recovery showing WT1 and KO1 continue to be negative for β-catenin in hepithelial cells (200× ).

**Figure supplement 2.** Immunohistochemistry for glutamine synthetase (GS) in WT1 and KO1 mice at 6 months of recovery showing KO1 continue to be negative for GS as there is no cell source for β-catenin+ hepatocytes in this model (100× ).

previously (*Figure 4—figure supplement 3D*; *Pradhan-Sundd, 2018b*). Thus, intact AJs are present in both models during recovery and cannot be the basis of sustained DR and fibrosis in KO1.

## Prolonged periportal inflammation in KO1 but not in KO2 mice during recovery from CDE diet correlates with ongoing DR and fibrosis

Previously, the presence of immune cell infiltration has been shown to be essential in the development of DR and fibrosis. Mice lacking Th1 immune signaling or interferon-γ showed impaired DR and decreased fibrosis after CDE diet (*Knight, 2007*). To address inflammation, we performed staining for CD45, a pan-leukocyte marker. There were high numbers of CD45-positive cells in WT2 and KO2 mice after 2 -week CDE diet, which declined in WT2 after 2 weeks of recovery, decreased overall but persisted pan-lobularly in KO2 up to 3 months, and normalized to WT2 levels by 6 months (*Figure 5A*). CD45-positive cells were present in high numbers in both WT1 and KO1 at 2 weeks of CDE diet, and while these numbers gradually returned to baseline in WT2 at 2 weeks of recovery, a more intense periportal-appearing infiltration was seen in KO1 especially at 3 months and 6 months (*Figure 5B*). Quantification of the area occupied by CD45-positive cells at all time points revealed that CD45-positive cell infiltration significantly resolved at 3 months and 6 months of recovery from CDE diet in both WT2 and KO2 (*Figure 5—figure supplement 1A*). However, CD45-positive cells in KO1 always tended to be higher than WT1 at all respective time points analyzed. Statistically significant and higher inflammation was observed in KO1 at 6 months as compared to WT1 at all time points (*Figure 5—figure supplement 1B*).

To determine if these inflammatory cells were close to DR, we performed triple immunofluorescence (IF) for PanCK, CD45, and myofibroblast marker α-smooth muscle actin (αSMA) (*Figure 5—figure supplement 2A*). At 2- week CDE diet, both WT2 and KO2 livers exhibited inflammatory cells and αSMA-positive cells close to BECs. After 2 weeks of recovery, αSMA-positive cells were no longer associated with BECs in WT2 and no immune cells were seen. In KO2 mice, BECs, αSMA-positive cells, and leukocytes were seen in close proximity to each other up to 3 months of recovery and returned to WT2 state at 6 months (*Figure 5—figure supplement 2A*). KO1 livers showed closely associated CD45- and PanCK-positive cells at all times in contrast to WT1, which lacked CD45 cells at all times after 2 weeks of recovery (*Figure 5—figure supplement 2A*). Intriguingly, even in KO1, αSMA-positive cells were not observed at any time after 2 weeks of recovery. Overall, the presence of CD45-positive cells close to PanCK-positive cells was clear in KO1 but not in KO2 at 6 months (*Figure 5C*).

Analysis in whole livers for expression of *Adgre1*, gene encoding macrophage marker F4/80, showed significant increase in KO1 compared to WT1 mice at 3- month recovery (*Figure 5D*), and these macrophages were located close to the DR (*Figure 5—figure supplement 2B*). Bone marrow monocyte-derived macrophages express high levels of *Itgam* (CD11b), and infiltrate liver during injury, express pro-inflammatory cytokines, and are involved in both progression and recovery phases of fibrosis (*Guillot and Tacke, 2019*). *Itgam* expression was significantly higher in KO1 at 3- month and 6 -month recovery times (*Figure 5D*). There was also increased staining for Ly6G, a marker for monocytes, granulocytes, and neutrophils, in KO1 at 3 -month and 6 -month recovery times (*Figure 5E*).

Together, these results show resolution of DR and fibrosis in KO2 correlated with reduced inflammation, whereas persistent β-catenin-negative DR and continuing fibrosis in KO1 mice at late recovery stages from CDE injury were associated with persistent periportal inflammation.

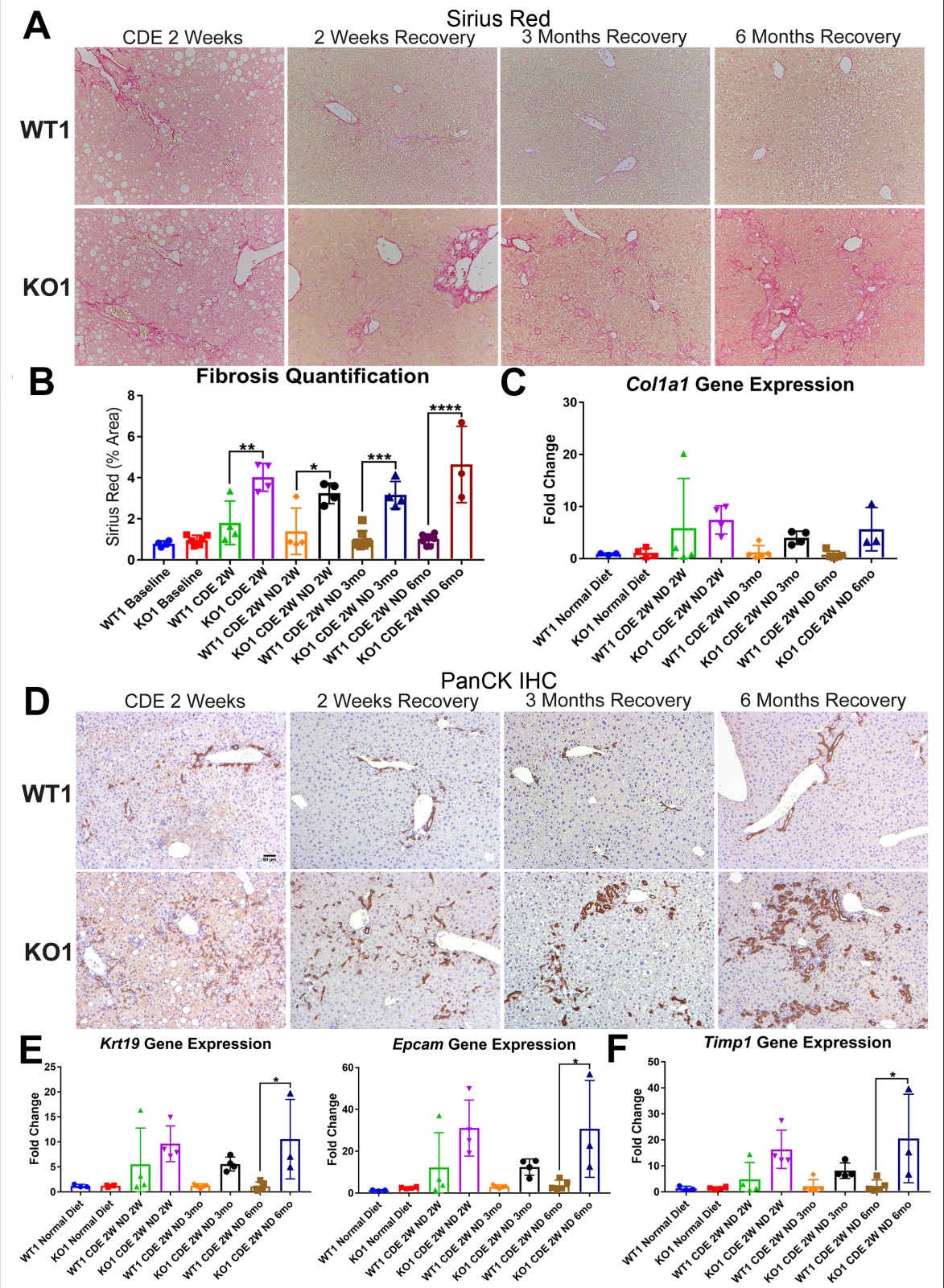

**Figure 4.** Unresolved fibrosis and ductular reaction in KO1 mice throughout 6 months on recovery, after the initial 2- week choline-deficient, ethionine-supplemented (CDE) diet injury. (**A**) Sirius Red staining in WT1 and KO1 mice over time during recovery from CDE diet (100× ). (**B**) Quantification of Sirius Red staining (one-way ANOVA, *p<0.05, **p<0.01, ***p<0.001, ****p<0.0001, n = 3–6 per group, individual animal values represented by dots). (**C**) A trend of increased expression of *Col1a1* in KO1 mice even at 6 months of recovery (n = 3–5 per group, individual animal values represented by dots).

*Figure 4 continued on next page*

Figure 4 continued

(**D**) Pan-cytokeratin (PanCK) staining in WT1 and KO1 mice over time during recovery from CDE diet. The ductular reaction (DR) changes from flattened, invasive. and without lumen morphology from early time points to numerous small luminal structures lined by a single layer of PanCK-positive columnar cells at 3 months and 6 months. Scale bar = 50 μm. (**E**) Significantly higher *Krt19* and *Epcam* gene expression in KO1 especially at 6 months of recovery (one-way ANOVA, *p<0.05, n = 3–5 per group, individual animal values represented by dots). (**F**) Significantly higher *Timp1* gene expression IN KO1 especially at 6 months of recovery diet (one-way ANOVA, *p<0.05, n = 3–5 per group, individual animal values represented by dots).

The online version of this article includes the following figure supplement(s) for figure 4:

**Figure supplement 1.** Enhanced ductular reaction (DR) during recovery from choline-deficient, ethionine-supplemented (CDE) in KO1.

**Figure supplement 2.** Enhanced proliferating cell nuclear antigen (PCNA) and increased p-Erk staining in biliary epithelial cells (BECs) in KO1 during recovery on normal diet after initial 2 -week choline-deficient, ethionine-supplemented (CDE) diet-induced injury.

**Figure supplement 3.** Bile acids and cell death are not the basis of fibrosis and ductular reaction due to choline-deficient, ethionine-supplemented (CDE) diet, and differences in adherens junction integrity do not explain phenotypic differences between KO2 and KO1 at 6 months of recovery.

**Figure supplement 3—source data 1.** Immunoprecipitation (IP) shows E-cadherin association with β-catenin in WT1, WT2, and KO2 livers at 6 months of recovery while it associates with γ-catenin in KO1 at the same time due to continued lack of β-catenin in KO1 (top panels).

## Lack of β-catenin from BECs in KO1 leads to persistent NF-κB activation during recovery from CDE diet-induced injury

To address the mechanism of enhanced periportal inflammation, we next assessed the status of NF-κB, the master regulator of immune cell response. We have previously shown an inhibitory inter-action of β-catenin with p65 subunit of NF-κB in hepatocytes and the absence of β-catenin in KO1 led to increased NF-κB activation in response to lipopolysaccharide (LPS) or tumor necrosis factor-α challenge (*Nejak-Bowen et al., 2013*). Further, while immune cells are essential for DR and fibrosis, reactive ductules are a well-known source of pro-inflammatory and pro-fibrogenic cytokines and thus this cross-cellular signaling perpetuates overall injury (*Fava et al., 2005*; *Pinto et al., 2018*). Since inflammatory cells were specifically enriched in KO1 in the periportal region and associated closely to the DR, we next assessed NF-κB status along with β-catenin in BECs in KO2 and KO1. At baseline in KO2, p65 subunit of NF-κB was present in the cytosol of the CK19-positive cells, as was β-catenin (*Figure 6A*). In KO1, at baseline, CK19-positive BECs lacked β-catenin and p65 was still evident in cytosol (*Figure 6A*). At 2 weeks of CDE diet, when DR and inflammation is ongoing in both KO2 and KO1 livers, a subset of CK19-positive BECs showed comparable nuclear p65 by confocal microscopy in both groups (*Figure 6B and C*). At 6 months in KO2, β-catenin-positive, CK19-positive BECs showed only cytosolic p65 similar to baseline (*Figure 6B*). However, unlike at baseline, KO1 livers at 6- month recovery from CDE injury showed profound nuclear translocation of p65 in almost all CK19-positive BECs, which continued to lack β-catenin (*Figure 6B*). Quantification showed significant difference in nuclear p65 in BECs in KO1 versus KO2 at 6 -month recovery time point (*Figure 6C*).

To verify if nuclear p65 indicated NF-κB activation, 84 downstream target genes were checked by RT-PCR array (fold-change threshold = >2, p-value threshold = 0.05). Relative to WT1, we found a striking upregulation in the expression of 44% of genes (37/84) in KO1, whereas 92% of genes (77/84) in KO2 livers were unchanged at 6 months of recovery (*Figure 6E*). Clustergram showed KO2s were indistinguishable from WT1, but KO1 clearly separated from both groups (*Figure 6—figure supplement 1*).

Altogether, these data suggest a pronounced and prolonged NF-κB activation in BECs lacking β-catenin along with periportal inflammation during recovery phase from CDE injury, whereas pres-ence of β-catenin dampened NF-κB activation in BECs to curbed inflammation and assist in recovery from the same injury.

## β-Catenin modulation in BECs leads to differential impact on NF-κB activity and pro-inflammatory cytokine expression through alterations in β-catenin-P65 interactions

To more conclusively address the relationship between β-catenin and NF-κB in BECs directly, we utilized immortalized mouse small cholangiocyte cells (SMCCs), which were transfected with control- or β-catenin siRNA together with either β-catenin-TCF TOPFlash reporter or p65 luciferase reporter. Knockdown of β-catenin in SMCCs, shown by a significant decrease in TOPFlash activity, induced p65 luciferase activity, which was three times greater than caused by 100 ng/ml LPS stimulation

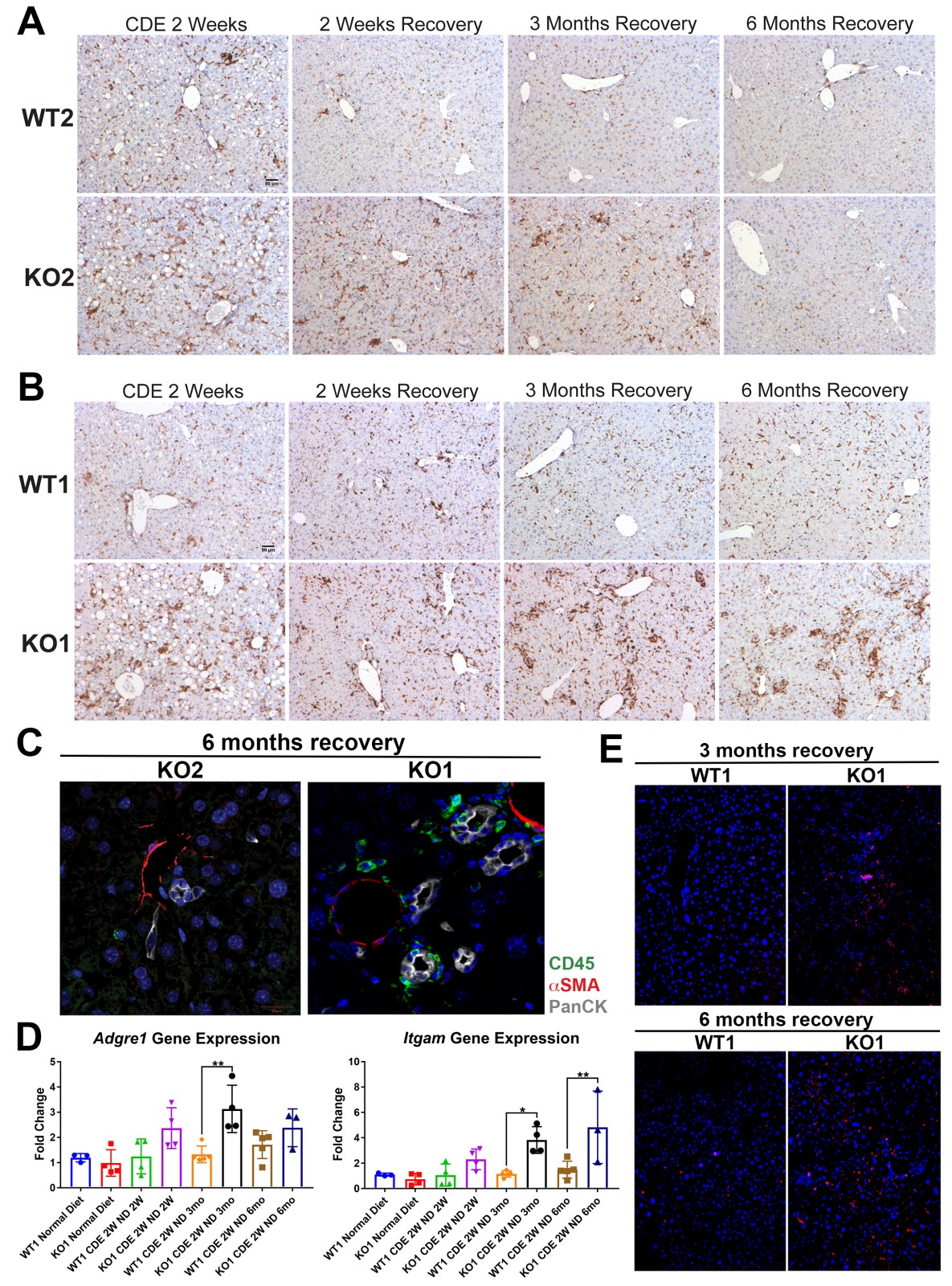

**Figure 5.** Sustained inflammation during recovery after the initial choline-deficient, ethionine-supplemented (CDE) diet injury in KO1 mice as compared to WT1, WT2, and KO2. (**A**) CD45 immunostaining in WT2 and KO2 mice over time during recovery from CDE diet. Scale bar = 50 μm. (**B**) CD45 staining in WT1 and KO1 mice over time during recovery from CDE diet. Scale bar = 50 μm. (**C**) Representative confocal image of triple immunofluorescence for pan-cytokeratin (PanCK) (white), α-smooth muscle actin (αSMA) (red), and CD45 (green) in KO1 and KO2 at 6 months of recovery (400× ). (**D**) Changes in

*Figure 5 continued on next page*

Figure 5 continued

*Adgre1* and *Itgam* gene expression in WT1 versus KO1 at various time points (one-way ANOVA, *p<0.05, **p<0.01, n = 3–5 per group, individual animal values represented by dots). (**E**) Increased Ly6G staining (red) in KO1 mice after 3 months and 6 months of recovery (100× ).

The online version of this article includes the following figure supplement(s) for figure 5:

**Figure supplement 1.** Increased CD45-positive cell infiltration in KO1 especially at 6 months of recovery from 2- week choline-deficient, ethionine-supplemented (CDE) injury.

**Figure supplement 2.** Immune cells continue to prevail in periportal region in KO1 even at 6 months of recovery while they subside in all other genotypes after 3 months of recovery or earlier.

(*Figure 7A*). Conversely, expression of stable S45Y-β-catenin or T41A-β-catenin (not shown) in SMCCs led to increased TOPFlash activity but significantly decreased p65 reporter activity with or without LPS (*Figure 7B*). Similar negative regulation was also observed in MzChA and HuCCT1, two independent human CCA cell lines (*Figure 7—figure supplement 1A and B*).

To address how β-catenin regulates NF-κB activity, SMCCs transfected with control- or Ctnnb1-siRNA were cultured with or without LPS, and subjected to cell fractionation to isolate nuclear- and cytoplasmic-enriched fractions. By WB, at baseline, notably higher levels of both p65 and β-catenin were evident in the cytoplasmic and not nuclear fraction. Upon β-catenin silencing, significantly higher levels of p65 were found in the nuclear compartment versus the controls, both without LPS, but more so following LPS treatment (*Figure 7C and D*).

Next, we modulated β-catenin activity in SMCCs to determine changes in global gene expression. Bulk RNA-seq was performed on β-catenin-silenced or control SMCCs, and on SMCCs transfected with S45Y-β-catenin or eGFP. Using a cutoff p-value ≤ 0.05 and abs(log2FC) ≥ 1.5, we found 335 β-catenin-regulated genes in BECs. Specifically, β-catenin-silenced SMCCs showed 75 upregulated and 122 downregulated, and β-catenin-active SMCCs showed 76 upregulated and 69 downregulated genes (*Figure 7—figure supplement 1C–E*). While there was a minimal overlap (*Figure 7—figure supplement 1E*), JASPAR was queried to identify transcription factor (TF) binding profiles in the 335 differentially expressed genes (DEGs). RELA (p65) was identified among the top TFs (ranking by p-value), with 9.6% of DEGs showing known RELA regulation (32/335, p=0.015) and 15.2% (51/335, p=0.088) showing NF-κB binding sites (*Figure 7E*). Interestingly, from RNA-seq and by qPCR, we found modest but significant increase in CCL2, and a more pronounced and significant increase in CXCL5 expression, in the β-catenin-silenced SMCCs (*Figure 7F*). After 6 -month recovery from CDE diet, a significant induction in CCL2 and CXCL5 expression was noted in KO1 but not in WT1 and KO2 (*Figure 7G*).

Next, we investigated if β-catenin interacts with p65 in BECs. Whole-cell lysates from SMCCs and normal mouse liver were used to pull down p65. We identified robust β-catenin association with p65 in SMCCs (*Figure 7H1*). A fainter but positive β-catenin-p65 association was evident in whole livers, likely due to low BEC representation in protein lysates from whole livers (*Figure 7I*). To further verify presence of β-catenin-p65 complex in BECs in vivo, we examined β-catenin and p65 localization using confocal microscopy. In WT1 liver at baseline, a notable colocalization of p65 and β-catenin was evident in the cytoplasm of BECs, which was expectedly absent in KO1 (*Figure 7—figure supplement 1F*). Quantification of colocalization using ImageJ showed that about 35% of p65 is associated with β-catenin, which was significantly greater than KO1 (*Figure 7J*).

Altogether, biochemical and IF studies identify a heretofore undescribed β-catenin-p65 complex in the cytoplasm of BECs. Further, β-catenin seems to prevent p65 nuclear translocation and NF-κB activation, and may be important in shutting off NF-κB activation when its signaling is no longer required. Absence of β-catenin in BECs enhances nuclear translocation of p65 at baseline, and more so in the presence of known positive regulators of NF-κB signaling such as LPS, and contributes to enhanced expression of genes involved in proliferation, inflammation, and fibrosis (*Kim et al., 2015a*; *Luedde and Schwabe, 2011*).

## Nuclear P65 in pathological DR in subset of clinical cases divulges heretofore unidentified interactions of β-catenin, P65, and CFTR

Since persistent DR observed in KO1 at 3 months and 6 months after recovery from CDE injury showed unique morphology consisting of numerous small luminal structures lined by a single layer of

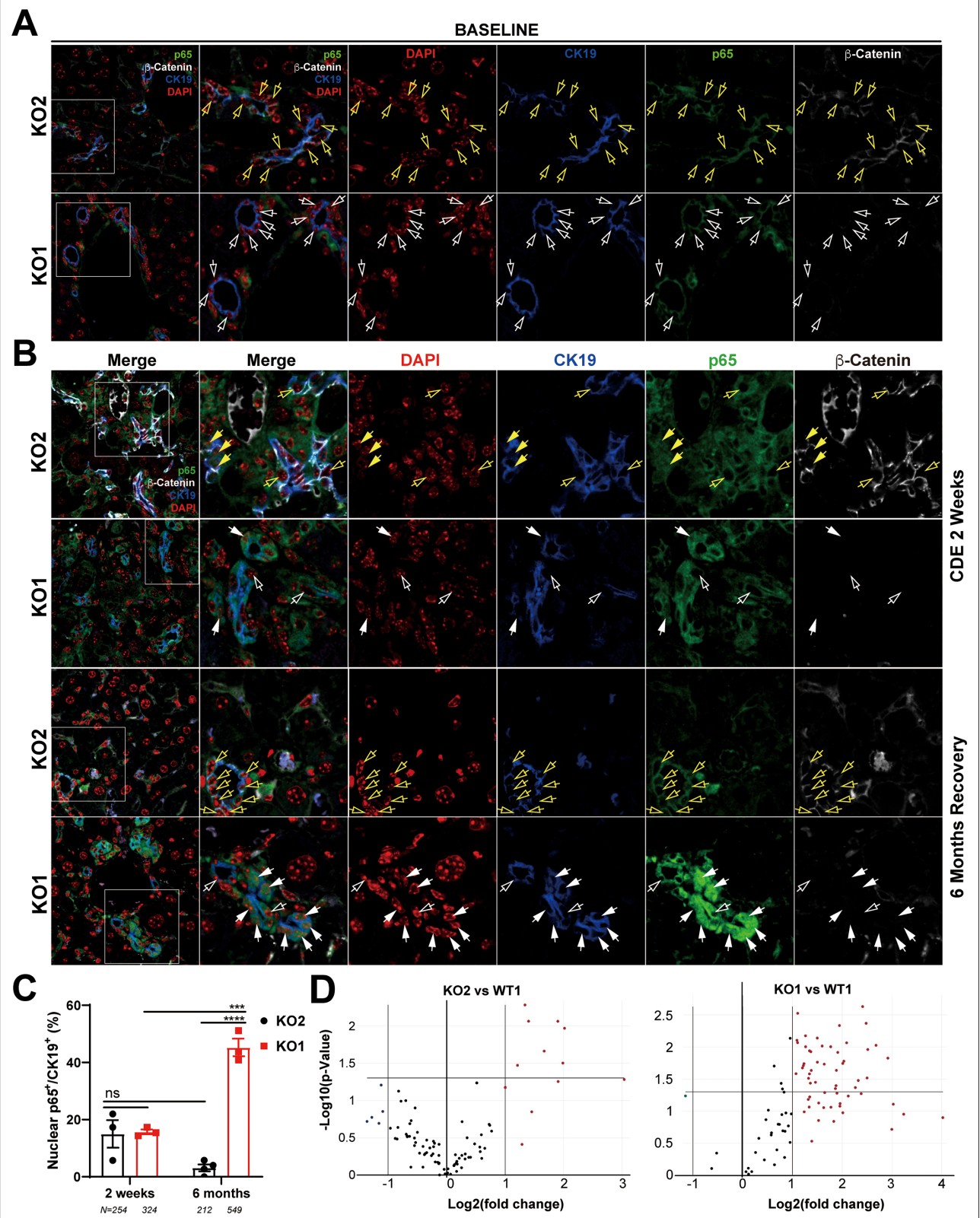

**Figure 6.** Nuclear translocation of p65 in biliary epithelial cells (BECs) lacking β-catenin during recovery from choline-deficient, ethionine-supplemented (CDE) diet injury shows NF-κB activation in BECs only in KO1. (**A**) Representative confocal image of triple immunofluorescence for CK19 (blue), β-catenin (white), and p65 (green) along with DAPI (red) in KO1 and KO2 baseline livers. Merged image at low magnification (100×) is shown in leftmost panel and higher magnification (200×) of selected area (box) along with its individual channels are shown to the right. Yellow open arrows identify

*Figure 6 continued on next page*

*Figure 6 continued*

CK19-positive BECs with cytosolic β-catenin and p65 in KO2. White open arrows identify CK19-positive BECs with cytosolic p65 and absent β-catenin in KO1. (**B**) Representative confocal image of triple immunofluorescence for CK19 (blue), β-catenin (white), and p65 (green) along with DAPI (red) in KO1 and KO2 at 2 weeks of CDE diet and after 6 months of recovery on normal diet. The leftmost panel is low magnification (100× ) merged image. The higher magnification (200× ) of the selected boxed area is presented in the adjacent panel as a merged image followed by individual channels. Yellow open arrows identify CK19-positive BECs with cytosolic β-catenin and p65, and yellow solid arrows with nuclear p65 in KO2. White open arrows indicate CK19-positive BECs with cytosolic p65, and white solid arrows identify CK19-positive BECs with nuclear p65 with absent β-catenin in KO1. (**C**) Quantification of CK19-positive cells showing nuclear p65 at 2 weeks of CDE diet and 6 months of recovery in KO2 versus KO1 (one-way ANOVA, ***p<0.001, ****p<0.0001, n = 3 per group, individual animal values represented by dots, number of cells counted are indicated). (**D**) Volcano plots of NF-$\kappa$B downstream target gene expression in KO2, KO1, and WT1 livers at 6 months of recovery. Genes with fold-change >2 are highlighted in red, with fold-change <2 highlighted in green, and unchanged genes shown as black dots (n = 3 per group).

The online version of this article includes the following figure supplement(s) for figure 6:

**Figure supplement 1.** Evidence of NF-$\kappa$B activation in KO1 livers but not in WT1 or KO2 livers at 6 months of recovery from choline-deficient, ethionine-supplemented (CDE) diet.

PanCK-positive cells, we decided to interrogate livers from clinical cases exhibiting DR including alcoholic hepatitis (AH), polycystic liver disease (PLD), and CF. As seen by panCK IF, AH cases displayed DR with variable morphology including areas of luminal small DR (shown in representative case), whereas PLD showed large cysts lined by flattened BECs (*Figure 8A*). DR in CF was more homogeneous and appeared uniformly reminiscent of what we observed in 3 -month and 6- month recovery times in KO1 (*Figures 4D and 8A*). Intriguingly, strongest and significant nuclear p65 was consistently observed in DR seen in CF cases followed by PLD with only a very small subset of cells in DR showing nuclear p65 in AH (*Figure 8B*). While β-catenin seem to be unaltered by IF staining in all three pathologies (*Figure 8A*), we observed a decrease in total β-catenin in a single CF case from whom two independent frozen liver samples were available (*Figure 8C*).

CF cases typically have varying loss-of-function (LOF) mutations in *CFTR* gene. Since β-catenin-p65 interactions were observed in SMCCs, we next assessed if CFTR is interacting with this complex. We observed concomitant pulldown of both B-band (faster migrating core glycosylated immature form) as well as slowly migrating C-band (complex glycosylated form) (*Chang, 2008*), along with p65, when we immunoprecipitated β-catenin in SMCCs (*Figure 8D*). To mimic LOF of CFTR seen in CF, we next silenced *Cftr* in SMCCs. Knockdown of *Cftr* led to a pronounced increase in p65 reporter activity (*Figure 8E*). Likewise, the expression of NF-κB target chemokines *Ccl2* and *Cxcl5* was significantly induced upon *Cftr* knockdown (*Figure 8F*). To query impact of β-catenin modulation, we next coexpressed stable-β-catenin (S45Y-mutant) in control and *Cftr*-siRNA-transfected SMCCs. Stabilization of β-catenin significantly decreased CFTR knockdown-induced p65 activation (*Figure 8E*).

To query if β-catenin loss in BECs had any impact on CFTR levels, we examined total levels of CFTR in KO1 and WT1 at baseline and at 6 months of recovery from CDE diet. We did not observe any differences in the levels of total CFTR protein, suggesting that CFTR may be an upstream effector of β-catenin while β-catenin does not regulate CFTR levels in BECs (*Figure 8—figure supplement 1*).

Thus, we identify important interactions between β-catenin, p65, and CFTR in BECs, and LOF of CFTR leads to destabilization of β-catenin, which in turn allows p65/NF-κB activation more readily in response to the available cues including the presence of cytokines. Also, β-catenin stabilization in BECs could tamper such enhanced p65 activation and may have potential therapeutic benefit in controlling unchecked NF-κB activation, inflammation, and fibrosis in reactive ductular cells.

## Discussion

DR is a common hallmark of many chronic liver pathologies, although its morphology is heterogeneous ranging from isolated invasive ductular cells, luminal phenotype, and sometimes purely cystic (*Sato et al., 2019*; *Wilson and Rudnick, 2019*; *Nejak-Bowen, 2020*). The significance of DR remains controversial and has been associated with both repair and disease progression (*Sato et al., 2019*; *Kamimoto et al., 2020*). Its role as a source of de novo hepatocytes through the process of transdifferentiation is indisputable in preclinical models shown by many fate-tracing studies (*Russell, 2019*; *Wei-Yu et al., 2015*; *Nejak-Bowen, 2020*; *Raven, 2017*). At the same time, DR can induce fibrosis by secreting pro-inflammatory and pro-fibrogenic factors to contribute to the disease process (*Lowes*

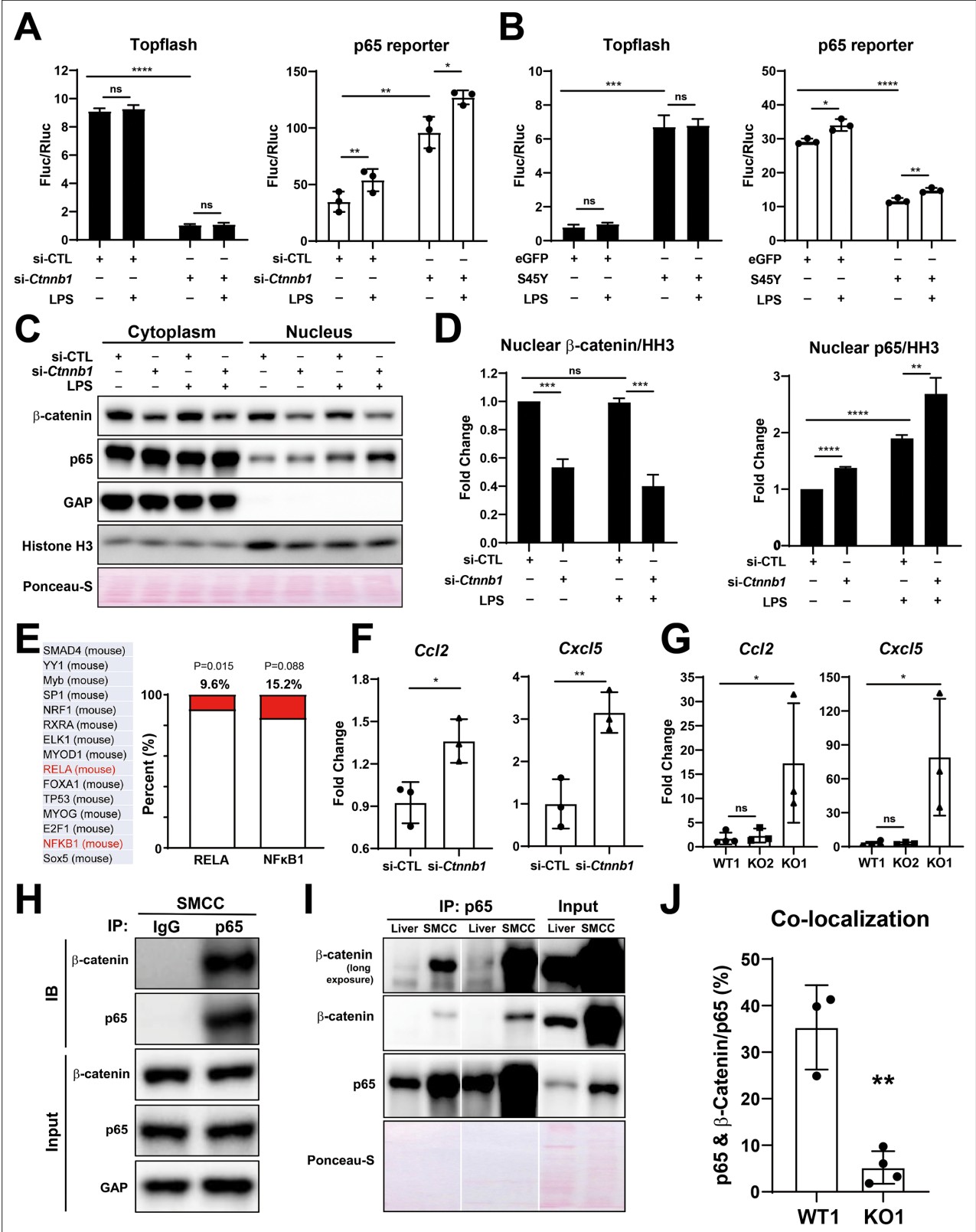

**Figure 7.** Modulation of β-catenin in biliary epithelial cells (BECs) perturbs its complex with p65 to impact NF-$\kappa$B activity. (**A**) Luciferase reporter assay shows successful knockdown of *Ctnnb1* in small cholangiocyte cell (SMCC) line by TOPFlash assay (left), which stimulates p65 transcriptional activity with or without 100 ng/ml lipopolysaccharide (LPS) (right) (unpaired t-test, ns: no significance, *p<0.05, **p<0.01, ****p<0.0001, n = 3 biological replication). (**B**) Luciferase reporter assay shows expression of constitutively active S45Y-β-catenin enhances TOPFlash (left) and suppresses p65 transcriptional

*Figure 7 continued on next page*

*Figure 7 continued*

activity with or without 100 ng/ml LPS (right) (unpaired t-test, ns: no significance, *p<0.05, **p<0.01, ***p<0.001, ****p<0.0001, n = 3 biological replication). (**C**) Representative WB from two independent experiment shows knockdown of *Ctnnb1* increases p65 nuclear translocation with or without 500 ng/ml LPS. (**D**) Quantification of nuclear β-catenin (left) and nuclear p65 (right) to HH3 (blots in (**C**) were technically quantified three times and p-value was calculated using unpaired t-test, ns: no significance, **p<0.01, ***p<0.001, ****p<0.0001). (**E**) Identification of RELA and NFKB1 among the top 15 transcription factors identified by applying the 335 differentially expressed genes (DEGs) to JASPAR. (**F**) qPCR shows knockdown of *Ctnnb1* in SMCCs induces *Ccl2* (left) and *Cxcl5* (right) expression (unpaired t-test, ns: no significance, *p<0.05, **p<0.01, n = 3 biological replication). (**G**) qPCR shows *Ccl2* (left) and *Cxcl5* (right) are induced in KO1 after 6 -month recovery of choline-deficient, ethionine-supplemented (CDE) diet (unpaired t-test, *p< 0.05, n = 3–4 biological replication). (**H**) Representative immunoprecipitation (IP) image from two independent experiment shows p65 is strongly associated with β-catenin in SMCC. (**I**) IP shows that p65 is associated with β-catenin in whole liver lysate (L: liver; S: SMCC; P: equal amount of liver and SMCC lysate). (**J**) Quantification of colocalization of p65 and β-catenin is significantly diminished in KO1 compared to WT1 (unpaired t-test, **p<0.01, n = 3–4 biological replication).

The online version of this article includes the following figure supplement(s) for figure 7:

**Source data 1.** WB shows knockdown of *Ctnnb1* increases p65 nuclear translocation with or without 500 ng/ml lipopolysaccharide (LPS).

**Source data 2.** Immunoprecipitation (IP) image shows p65 is strongly associated with β-catenin in small cholangiocyte cell (SMCC).

**Source data 3.** Immunoprecipitation (IP) shows that p65 is associated with β-catenin in whole liver lysate.

**Figure supplement 1.** Modulation of β-catenin in cholangiocytes impacts NF-κB activity due to p65-β-catenin complex.

*et al., 1999*; *Richardson, 2007*; *Zhao et al., 2018*; *Kim et al., 2015a*; *Aguilar-Bravo, 2019*). What drives the pro-inflammatory and pro-fibrogenic phenotype of these reactive BECs and what reverts these cells back to normal is poorly understood.

Previously, we and others have described β-catenin-p65 complex in hepatocytes, breast and colon cancer cells, which could inhibit NF-κB activation (*Nejak-Bowen et al., 2013*; *Deng, 2002*). Indeed, being a 'sticky' protein, β-catenin interacts with many proteins in a cell to modulate their activities (*Russell and Monga, 2017*). The exact biological significance of β-catenin-p65 interaction is not well understood and likely context dependent. We identify the existence of this complex in normal BECs. We show β-catenin-p65 complex to be present in the cytoplasm in the BECs to prevent nuclear translocation and activation of p65 and in turn keep NF-κB activity in check. While knockdown of β-catenin in BECs in vitro led to some baseline nuclear translocation, it allowed more profound nuclear translocation and activation of p65 in the presence of LPS, a well-known NF-κB activator. Conversely, β-catenin stabilization in BECs dampened NF-κB activation basally or in the presence of LPS. β-Catenin activation is observed in BECs within the DR during chronic liver injuries (*Apte et al., 2008*; *Hu, 2007*; *Okabe, 2016*). However, unlike in hepatocytes, β-catenin activation in BECs does not play a role in their proliferation (*Russell, 2019*; *Okabe, 2016*). Our current study shows that β-catenin stabilization in BECs may in fact be 'mopping' up p65 to dampen and eventually shutoff NF-κB activation, reverting BECs to their quiescence. NF-κB activation has been shown to be important in BEC proliferation and DR by regulating Jagged/Notch signaling (*Kim et al., 2015a*). And NF-κB in BECs has been suggested to play a role in inducing pro-inflammatory and pro-fibrogenic milieu as well through regulating expression of many of its target genes including *Ccl2* and *Cxcl5* (*O'Hara et al., 2013*). In summary, absence of β-catenin in BECs as seen in KO1 prevents β-catenin-p65 complex formation, leading to sustained NF-κB activation after injury, whereas this complex reforms in KO2 especially due to known β-catenin increase in BECs after injury, thus preventing chronic NF-κB activation. Molecular underpinnings of β-catenin stabilization in BECs especially during cholestatic liver injuries remain under investigation, although portal fibroblasts, macrophages, hepatocytes, and BECs have all been shown to secrete ligands like Wnt7a, Wnt7b, Wnt10a, and Wnt5a (*Hu, 2007*; *Okabe, 2016*; *Wilson, 2020*).

Another intriguing observation was the distinct morphology of DR evident in the recovery phase from the CDE injury in β-catenin-deficient livers, which was reminiscent of the histology of the CF cases (*Sakiani et al., 2019*). Surprisingly, very few BECs in AH showed nuclear p65, while PLD cases showed variable but increased nuclear p65 in cells lining the cysts. However, DR in CF cases exhibited strongest and consistent nuclear p65. This led us to investigate interactions between β-catenin-p65 and CFTR, whose gene is mutated in CF. CFTR protein is present in BECs only in the liver (*Cohn et al., 1993*). We observed a pulldown of CFTR (B-band and C-band) with β-catenin in a BEC line. To mimic LOF, which is the common end result of *CFTR* mutations in CF patients, we silenced *CFTR* in SMCC line, which led to a profound p65 activation that was decreased upon β-catenin stabilization. This

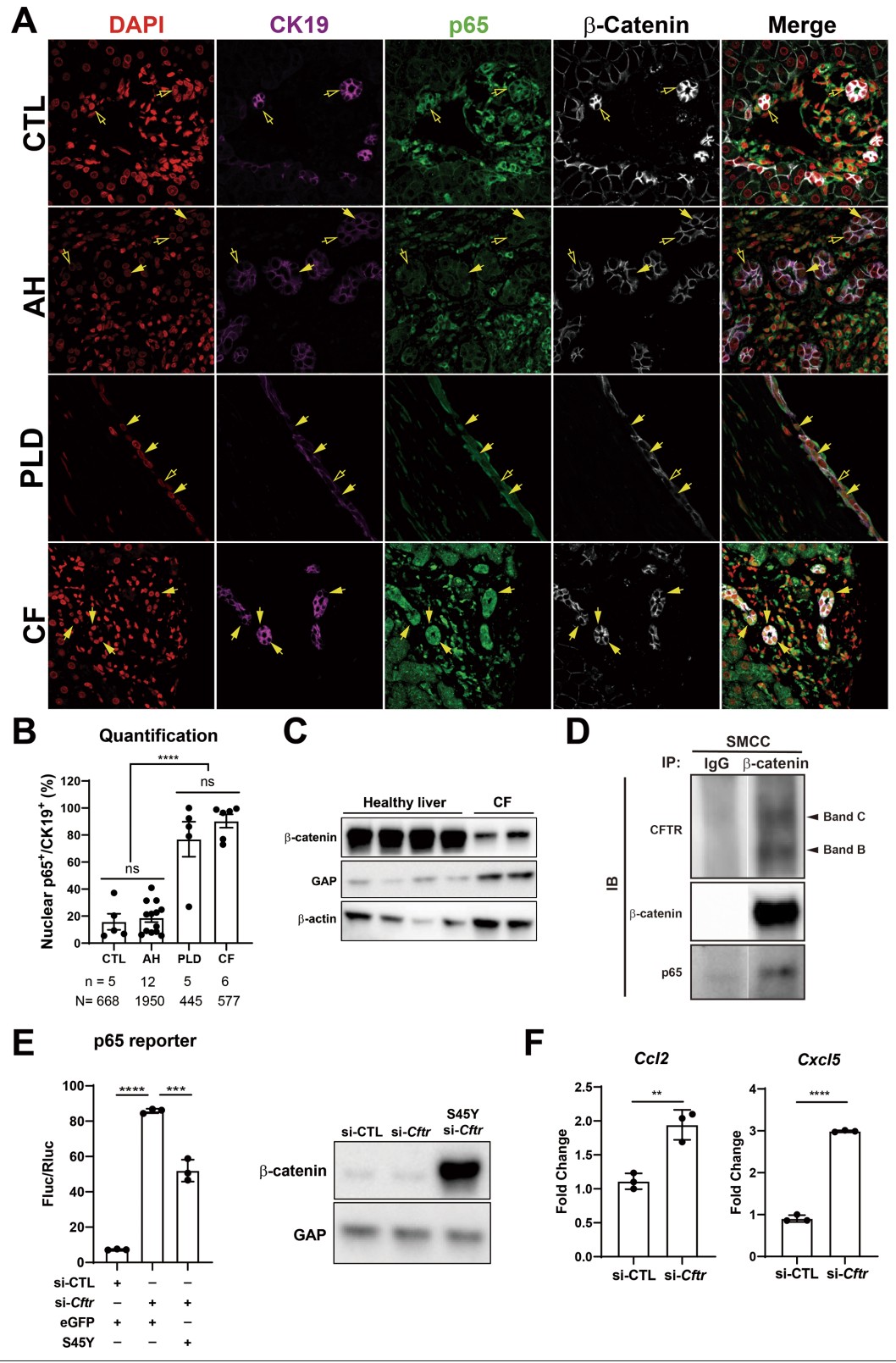

**Figure 8.** Nuclear p65 is highly present in the ductular cells of cystic type liver disease. (**A**) Representative immunofluorescence (IF) images of liver sections from patients with healthy liver (CTL), alcoholic hepatitis (AH), polycystic liver disease (PLD), and cystic fibrosis (CF) (200×). (**B**) Quantification of the percentage of nuclear p65+ CK19+ cells among CK19+ biliary epithelial cells (BECs) from CTL (668 cells from 5 patients), AH (1950 cells from

*Figure 8 continued on next page*

*Figure 8 continued*

12 patients), PF (445 cells from 5 patients), and CF (577 cells from 6 patients) (one-way ANOVA, ****p<0.0001). (**C**) WB shows β-catenin is decreased in two liver samples from one CF patient as compared to four healthy liver controls. (**D**) Immunoprecipitation (IP) studies show cystic fibrosis transmembranous conductance regulator (CFTR) (C-band and B-band) and p65 to be pulled down with β-catenin and not with IgG control in small cholangiocyte cell (SMCC). Images are from the same gel with same exposure time. (**E**) Luciferase reporter assay shows knockdown of CFTR in SMCC line strongly induces p65 transcriptional activity. Overexpression of stable S45Y-β-catenin partially rescues p65 activity (unpaired t-test, ***p<0.001, ****p<0.0001, n = 3 biological replication). Representative WB shows β-catenin levels after si-*Cftr* or after simultaneous S45Y-β-catenin expression as compared to si-control (si-CTL). (**F**) qPCR shows knockdown of CFTR induces *Ccl2* and *Cxcl5* expression in SMCC (unpaired t-test, **p<0.01, ***p<0.001, n = 3 biological replication).

The online version of this article includes the following source data and figure supplement(s) for figure 8:

**Source data 1.** WB shows β-catenin is decreased in two liver samples from one cystic fibrosis (CF) patient as compared to four healthy liver controls.

**Source data 2.** Immunoprecipitation (IP) studies show cystic fibrosis transmembranous conductance regulator (CFTR) (C-band and B-band) and p65 to be pulled down with β-catenin and not with IgG control in small cholangiocyte cell (SMCC).

**Source data 3.** WB shows β-catenin levels after si-*Cftr* or after simultaneous S45Y-β-catenin expression as compared to si-control (si-CTL).

**Figure supplement 1.** Total cystic fibrosis transmembranous conductance regulator (CFTR) levels are unaffected by loss of β-catenin in KO1 livers at baseline and during recovery from choline-deficient, ethionine-supplemented (CDE) injury.

**Figure supplement 1—source data 1.** Lack of changes in CFTR protein in whole liver lysates from KO1 livers.

suggests an important tripartite regulatory interaction between these proteins. Interestingly, in human lung epithelial cells, a proteomic screen identified interaction of β-catenin with WT CFTR but not with ΔF508 CFTR, a major site of LOF mutation in the *CFTR* (*Pankow, 2015*). Additionally, in mouse intestine, CFTR was shown to stabilize β-catenin and prevent its degradation (*Liu et al., 2016*). The same study showed that ΔF508 CFTR is unable to interact with β-catenin, leading to β-catenin degradation and eventually resulting in activation of NF-κB-mediated inflammatory cascade. Our results show an existence of a tripartite interaction between β-catenin, p65, and CFTR in cholangiocytes. LOF of CFTR or its reduced levels led to decrease in β-catenin protein in BECs both in vitro (SMCC) and in vivo (CF patient liver lysate), leading to nuclear translocation of p65, NF-κB activation, and increased expression of its pro-inflammatory chemokine targets. Absence of β-catenin in BECs however did not alter CFTR levels, suggesting CFTR to upstream effector of β-catenin while not being dependent on β-catenin for its own stability. We believe that the classical pathology of liver disease in CF including periductal inflammation, DR, periportal fibrosis, and focal biliary cirrhosis may be explained by our observations, in addition to previously described mechanisms such as Rous sarcoma oncogene cellular homolog (Src)-dependent Toll-like receptor-4 activation (*Fiorotto, 2016*). Since β-catenin activation in BECs inhibited NF-κB activation occurring due to CFTR silencing, this strategy may have therapeutic implications in controlling CF disease progression in the liver and elsewhere and future studies will directly address this novelty. NF-κB activation has been shown to be important in BEC proliferation and DR by regulating Jagged/Notch signaling (*Kim et al., 2015a*). The same study showed NF-κB activation to be regulated by cysteine-rich protein 61 (CYR61), whose knockdown reduced DR. Incidentally, *CFTR* was also significantly reduced in that study and might have been a mechanism of DR through disruption of CTFR-p65-β-catenin interactions. The mechanism of NF-κB activation and whether these tripartite interactions are playing any role in other pathologies such as PLD or subset of AH cases requires further investigation.

There is very little understanding of the process of resolution of DR along with its associated fibrosis, although these are strongly linked to inflammation (*Ko et al., 2019*). Our study provides novel insight into not only the molecular underpinnings of a reactive cholangiocyte, but also sheds light on how BECs regulate the immune microenvironment. Levels of Timp1 correlate with fibrosis, and several groups have investigated the use of TIMP1 levels as a biomarker for fibrosis in hepatitis C patients (*Leroy, 2004*; *Boeker et al., 2002*). Mice with overexpression of TIMP1 developed dramatically more fibrosis after CCl$_4$ treatment (*Yoshiji, 2000*), and TIMP1 transgenic mice showed impaired fibrosis

resolution after cessation of CCl$_4$ (*Yoshiji, 2002*). Our data is consistent since expression of *Timp1* was associated with fibrosis, and levels of *Timp1* normalized upon resolution of fibrosis.

# Materials and methods

## Animals

All animals are housed in temperature and light-controlled facilities and are maintained in accordance with the Guide for Care and Use of Laboratory Animals and the Animal Welfare Act. Generation of *Albumin-Cre;Ctnnb1^flox/flox^* mice and wild-type littermates has been described previously (*Tan et al., 2006*). Generation of *Ctnnb1^flox/flox^;Rosa-stop^flox/flox^-EYFP* reporter mice was also described previously (*Russell, 2019*). In brief, 23–25- day-old *Ctnnb1^flox/flox^;Rosa-stop^flox/flox^-EYFP* mice were injected intraperitoneally with 1 × 10$^{12}$ genome copies (GCs) of adeno-associated virus serotype 8 encoding Cre recombinase under the hepatocyte-specific thyroid binding globulin promoter (AAV8-TBG-Cre; Addgene) followed by a 12- day washout period. The same protocol was utilized on *Ctnnb1^+/+^;Rosa-stop^flox/flox^-EYFP* mice to generate WT2 mice. When mice were 4–6 weeks old, choline-deficient diet (Envigo Teklad Diets) supplemented with 0.15% ethionine drinking water (Acros Organics, 146170100) was administered for 2 weeks. For recovery time points, animals were switched back to normal chow diet for up to 6 months. Serum biochemistry analysis was performed by automated methods at the University of Pittsburgh Medical Center clinical chemistry laboratory. All studies were performed according to the guidelines of the National Institutes of Health and the University of Pittsburgh Institutional Animal Use and Care Committee.

## Patient data

All patient tissue sections were provided by the Pittsburgh Liver Research Center's (PLRC's) Clinical Biospecimen Repository and Processing Core (CBPRC), supported by P30DK120531. Sections from 5 patients with healthy liver, 12 patients with DR from AH (n = 10) and/or NASH (n = 2), 5 patients with DR associated with PLD, and 6 patients with DR in CF cases were triple stained with CK19, p65, and β-catenin for further analysis. Patient information from these groups of cases is listed in *Supplementary file 1*. Two pieces of frozen livers from one CF patient (TP10-P531) were provided by Pitt Biospecimen Core and used for WB. Information on this case is also included in *Supplementary file 1*.

## Immunohistochemistry

The IHC protocols have been described previously (*Russell, 2019*). In brief, liver tissue was fixed in 10% buffered formalin for 48 hr prior to paraffin embedding. Blocks were cut into 4 -μm sections, deparaffinized, and washed with PBS. For antigen retrieval, samples were microwaved for 12 min in pH 6 sodium citrate buffer (PanCK, CD45, p-Erk1/2, cleaved caspase 3) or Tris-EDTA buffer (p21), were pressure cooked for 20 min in pH 6 sodium citrate buffer (β-catenin), or were incubated with Proteinase K (Agilent Dako, S302030-2) for 10 min (F4/80). Samples were then placed in 3% H$_2$O$_2$ for 10 min to quench endogenous peroxide activity. After washing with PBS, slides were blocked with Super Block (ScyTek Laboratories, AAA500) for 10 min or 10% goat serum in PBS for 10 min (p21). The primary antibodies were incubated at the following concentrations in antibody diluent: PBS + 1% BSA (Fisher BioReagents, BP1605-100) with 0.1% Tween 20 (Fisher BioReagents, BP337-500): PanCK (Dako, Z0622, 1:200), cleaved caspase 3 (Cell Signaling, 9664, 1:100), p-Erk1/2 (Cell Signaling, 4370, 1:100), F4/80 (Bio-Rad, MCA497A488, 1:100) for 1 hr at room temperature or at 4 °C overnight: β-catenin (Abcam, ab32572, 1:50) and p21 (Santa Cruz, sc-471, 1:25). Samples were washed with PBS three times and incubated with the appropriate biotinylated secondary antibody (Vector Laboratories) diluted 1:500 in antibody diluent for 30 min at room temperature. Samples were washed with PBS three times and sensitized with the Vectastain ABC kit (Vector Laboratories, PK-6101) for 30 min. Following three washes with PBS, color was developed with DAB Peroxidase Substrate Kit (Vector Laboratories, SK-4100), followed by quenching in distilled water for 5 min. Slides were counterstained with hematoxylin (Thermo Scientific, 7211), dehydrated to xylene, and coverslips applied with Cytoseal XYL (Thermo Scientific, 8312-4). For Sirius Red staining, samples were deparaffinized and incubated for 1 hr in Picro-Sirius Red Stain (American MasterTech, STPSRPT), washed twice in 0.5% acetic acid water, dehydrated to xylene, and coverslipped. Images were taken on a Zeiss Axioskop 40 inverted brightfield microscope.

## Immunofluorescence

Liver tissue was fixed in 10% buffered formalin overnight, cryopreserved with 30% sucrose in PBS overnight, frozen in OCT compound (Sakura, 4583), and stored at –80 °C. OCT-embedded samples were cut into 5 -µm sections, allowed to air-dry, and then washed in PBS. Antigen retrieval was performed through microwaving in pH 6 sodium citrate buffer. Slides were washed with PBS and permeabilized with 0.1% Triton X-100 in PBS for 20 min at room temperature. Samples were washed three times with PBS and then blocked with 2% donkey serum in 0.1% Tween 20 in PBS (antibody diluent) for 30 min at room temperature. Antibodies were diluted as follows: PanCK (Dako Z0622, 1:200), PCNA (Santa Cruz Biotechnology, sc-56, 1:1000), p65 (Santa Cruz Biotechnology, sc-372, 1:500), β-catenin (BD Biosciences, 610154, 1:500), CK-19 (DSHB, TROMA-III, 1:10) in antibody diluent, and incubated at 4 °C overnight. Ly6G (clone: RB6-8C5) antibody was purchased from Thermo Fisher Scientific, Waltham, MA. Samples were washed three times in PBS and incubated with the proper fluorescent secondary antibody (AlexaFluor 488/555/647, Invitrogen) diluted 1:400 in antibody diluent for 2 hr at room temperature. The αSMA antibody is directly conjugated to Cy3 and requires no secondary antibody. Samples were washed three times with PBS and incubated with DAPI (Sigma, B2883) for 1 min. Samples were washed three times with PBS and mounted with fluoromount (SouthernBiotech) or ProLong Gold antifade reagent (Invitrogen, P10144). Images were taken on a Nikon Eclipse Ti epifluorescence microscope or a Zeiss LSM700 confocal microscope.

## Image quantification

To determine BEC proliferation, for each sample seven images at 200× magnification of periportal regions were taken, and in each image the number of PanCK+/PCNA+ cells was manually counted. For quantification of Sirius Red, PanCK, CD45 staining, staining intensity was measured in ImageJ for 3–5 images at 100× magnification per mice sample. Positive area to whole area ratio was calculated as percent-positive area using the NIH ImageJ software. To determine p65 nuclear-positive cholangiocytes, four images from each animal or patient at 200× magnification of DR regions were counted.

## RT-PCR

Whole liver was homogenized in TRIzol (Thermo Scientific, 15596026) and nucleic acid was isolated through phenol-chloroform extraction. Cellular DNA was digested with DNA-free Kit (Ambion, AM1906), and RNA was reverse-transcribed into cDNA using SuperScript III (Invitrogen, 18080-044). Real-time PCR was performed in technical triplicate on a StepOnePlus Real-Time PCR System (Applied Biosystems, 4376600) or on a Bio-Rad CFX96 Real-Time System using the Power SYBR Green PCR Master Mix (Applied Biosystems, 4367660). Target gene expression was normalized to the average of two housekeeping genes (*Gapdh* and *Rn18s*), and fold-change was calculated utilizing the ΔΔ-Ct method. For RT-PCR arrays, 2 µl of cDNA, 7.5 µl of Power SYBR Green PCR Master Mix, and 4.5 µl of nuclease-free water were premixed and added to each well of RT² Profiler PCR Array Mouse NFκB Signaling Pathway (Qiagen, PAMM-025Z) for target gene qPCR. RT-PCR arrays data were analyzed at GeneGlobe (https://geneglobe.qiagen.com/us/analyze/). The average of three housekeeping genes (*Rn18s, Actb,* and *Gapdh*) was used for normalization. Volcano plot and clustergram were generated by the data analysis web portal mentioned above.

## Immunoprecipitation and western blot

Whole liver tissue was homogenized in RIPA buffer premixed with fresh protease and phosphatase inhibitor cocktails. Cytoplasmic and nuclear extracts were prepared using the NE-PER Nuclear and Cytoplasmic Extraction Reagents (Thermo Fisher Scientific, 78835). The concentration of the protein was determined by the bicinchoninic acid assay. For IP, 1 mg of SMCC lysate was precleared with 40 µl of Protein A/G PLUS-Agarose (Santa Cruz Biotechnology, sc-2003) for 2 hr at 4 °C. After centrifugation (3000 rpm, 1 min), the supernatant was incubated with 2 µg of p65 antibody (Santa Cruz Biotechnology, sc-8008, 1:100), 2 µg of β-catenin antibody (BD Biosciences, 610154, 1:100), or control IgG overnight at 4 °C. The next day, samples were incubated with 40 µl of Protein A/G PLUS-Agarose for 1 hr at 4 °C. The pellet was collected, washed with RIPA buffer for three times, resuspended in 10 µl of loading buffer, and subjected to electrophoresis. Protein lysate was separated on pre-cast 7.5% or 4–20% polyacrylamide gels (Bio-Rad) and transferred to the PVDF membrane using the Trans-Blot Turbo Transfer System (Bio-Rad). Membranes were blocked for 75 min with 5% skim milk (Lab

Scientific, M0841) or 5% BSA in Blotto buffer (0.15 M NaCl, 0.02 M Tris pH 7.5, 0.1% Tween in dH2O), and incubated with primary antibodies at 4 °C overnight at the following concentrations: β-catenin (Cell Signaling, 8480, 1:1,000 in 1% BSA), p65 (Cell Signaling, 8242, 1:1000 in 1% BSA), CFTR (Alomone Labs, ACL-006, 1:250 in 1% BSA), GAPDH (Cell Signaling, 5174, 1:10,000 in 1% milk), and histone H3 (Cell Signaling, 9715, 1:1,000 in 1% milk). Membranes were washed in Blotto buffer and incubated with the appropriate HRP-conjugated secondary antibody for 75 min at room temperature. Membranes were washed with Blotto buffer, and bands were developed utilizing SuperSignal West Pico Chemiluminescent Substrate (Thermo Scientific, 34080) and visualized by autoradiography.

## Cell culture and reporter assays

SMCCs (obtained from Dr. Kari Nejak-Bowen, University of Pittsburgh, PA), and human CCA cell lines MzChA and HuCCT1 (obtained from Dr. Gregory J. Gores, Mayo Clinic, Rochester, MN) were seeded on six-well plates in a humidity-saturated incubator with 5% $CO_2$ maintained at 37 °C. Cells were mycoplasma negative tested with MycoAlert Mycoplasma Detection Kit (Lonaz, LT07-218). For p65 reporter assay, cells were transfected with 2 µg of p65 reporter and 0.2 µg of *Renilla* (internal control) together with 2 µg of either eGFP (control) or S45Y to overexpress constitutively active β-catenin using Lipofectamine 3000 Transfection Reagent (Invitrogen, L3000008), or together with si-Control (Cell Signaling, 6568) and si-β-catenin (Cell Signaling, 6225) to knock down β-catenin using Lipofectamine RNAiMAX Transfection Reagent (Invitrogen, 13778150). For TOPFlash reporter assay, p65 reporter above was replaced by TOPFlash plasmid. Cells were treated with 100 ng/ml of LPS 6 hr before harvest. si-CFTR (Santa Cruz Biotech, sc-35053) was used to knock down CFTR in SMCCs. Cells were harvested at 48 hr, and luciferase signals were got using Dual-Luciferase Reporter Assay System (Progema, E1910) and normalized to the value of *Renilla*.

## Measurement of hepatic bile acids

Total hepatic bile acids were measured using the Mouse Total Bile Acids Assay Kit from Crystal Chem (Downers Grove, IL), as per the manufacturer's instructions. To isolate total bile acids from liver, 50–100 mg frozen liver tissue was homogenized in 70% ethanol at room temperature, then samples were incubated in capped glass tubes at 50 °C for 2 hr. The homogenates were centrifuged at 6000 g for 10 min to collect the supernatant. Total bile acid concentrations were determined using the calibration curve from the standard provided in the kit and the mean change in absorbance value for each sample.

## RNA-seq and analysis

Twelve SMCC RNA samples were measured: three for CTNNB1 activation (SMCC-S45Y), three for CTNNB1 activation control (SMCC-eGFP), three for CTNNB1 silencing (SMCC-si-β-catenin), and three for CTNNB1 silencing control (SMCC-si-Control). In total, 12 RNA-seq libraries were sequenced. For each library, quality control was performed to each raw sequencing data by tool FastQC. Based on the QC results, low-quality reads and adapter sequences were filtered out by tool Trimmomatic (*Bolger et al., 2014*). Surviving reads were then aligned to mouse reference genome mm10 by aligner Hisat2 (*Kim et al., 2015b*). HTSeq tool (*Anders et al., 2015*) was then applied to the aligned file for gene quantification. Based on the gene count, differential expression analysis was applied to compare CTNNB1 activation samples with their corresponding controls, and to compare CTNNB1 silencing samples with their corresponding controls, respectively. R package DESeq2 (*Love et al., 2014*) was employed to perform the test, and DEGs were defined as genes with fold-change higher than 1.5-fold and p-value (or adjusted p-value) smaller than 0.05. These DEGs were further used to detect common upstream TFs based on the JASPAR database (*Fornes, 2020*). Opposite regulation directions of the activation and silencing models were finally compared in terms of DEGs. All statistical analyses were performed by R programming. Raw RNA-seq data and gene count quantification were submitted to NCBI GEO database with accession ID GSE155981 (https://www.ncbi.nlm.nih.gov/geo/query/acc.cgi? acc=GSE155981).

## Statistics

For analysis of serum biochemistry between two groups, a two-tailed t-test was performed. For analysis of cell counts, such as proliferating BECs, a Mann–Whitney U test was performed. A p<0.05 was

considered significant, and plots are mean ± SD. Detailed statistic information for each assay is given in the figure legends. All statistical analysis and graph generation were performed using GraphPad Prism software.

## Acknowledgements

This work was supported by NIH grants 1R01DK62277, 1R01DK100287, 1R01DK116993, R01CA204586, 1R01CA251155-01, and Endowed Chair for Experimental Pathology to SPM. This work was also supported in part by 1R01CA258449 to SK. This work was also supported by T32EB0010216 and 1F31DK115017-01 to JOR. This work was also supported by P30DK120531 to the Pittsburgh Liver Research Center for services provided by Biospecimen Repository and Processing Core.

## Additional information

### Funding

| Funder | Grant reference number | Author |
|---|---|---|
| National Institutes of Health | 1R01DK62277<br>1R01DK100287<br>1R01DK116993<br>R01CA204586<br>1R01CA251155-01 | Satdarshan P Monga |
| National Institutes of Health | 1R01CA258449 | Sungjin Ko |
| National Institutes of Health | T32EB0010216<br>1F31DK115017 | Jacquelyn O Russell |
| National Institutes of Health | P30DK120531 | Satdarshan P Monga |

The funders had no role in study design, data collection and interpretation, or the decision to submit the work for publication.

### Author contributions

Shikai Hu, Data curation, Formal analysis, Investigation, Methodology, Writing – original draft; Jacquelyn O Russell, Data curation, Investigation, Methodology, Writing – original draft; Silvia Liu, Data curation, Formal analysis, Software; Catherine Cao, Jackson McGaughey, Minakshi Poddar, Sucha Singh, Methodology; Ravi Rai, Data curation, Investigation, Methodology; Karis Kosar, Junyan Tao, Edward Hurley, Aaron Bell, Investigation, Methodology; Donghun Shin, Investigation, Resources; Reben Raeman, Formal analysis, Investigation, Resources; Aatur D Singhi, Formal analysis, Investigation, Resources, Visualization; Kari Nejak-Bowen, Conceptualization, Investigation, Resources; Sungjin Ko, Formal analysis, Funding acquisition, Investigation, Methodology, Project administration, Writing – review and editing; Satdarshan P Monga, Conceptualization, Formal analysis, Funding acquisition, Investigation, Project administration, Resources

### Author ORCIDs

Donghun Shin http://orcid.org/0000-0002-7975-9014
Satdarshan P Monga http://orcid.org/0000-0002-8437-3378

### Ethics

The study was conducted according to the guidelines of the Declaration of Helsinki, and approved by the Institutional Review Board of the University of Pittsburgh (STUDY19070068, STUDY20010114, and STUDY20040276 on 3/23/2021).

This study was performed in strict accordance with the recommendations in the Guide for the Care and Use of Laboratory Animals of the National Institutes of Health. All animals were handled according to approved institutional animal care and use committee (IACUC) Protocol #: 19126451 of the University of Pittsburgh.

Decision letter and Author response
Decision letter https://doi.org/10.7554/eLife.71310.sa1
Author response https://doi.org/10.7554/eLife.71310.sa2

## Additional files

### Supplementary files

• Supplementary file 1. Patient information whose samples were used in the study. Patient tissue sections including 5 patients with healthy liver, 12 patients with ductular reaction (DR) from alcoholic hepatitis (AH) (n = 10) and/or NASH (n = 2), 5 patients with DR associated with polycystic liver disease (PLD), and 6 patients with DR in cystic fibrosis (CF) cases were provided by the Pittsburgh Liver Research Center's (PLRC's) Clinical Biospecimen Repository and Processing Core (CBPRC). Two pieces of frozen livers from one CF patient (TP10-P531) were provided by Pitt Biospecimen Core.

• Transparent reporting form

### Data availability

Raw RNA-seq data and gene count quantification were submitted to NCBI GEO data base with accession ID GSE155981.

The following dataset was generated:

| Author(s) | Year | Dataset title | Dataset URL | Database and Identifier |
|---|---|---|---|---|
| Satdarshan PM | 2021 | β-Catenin-NFkB-CFTR interactions in cholangiocytes regulate inflammation and fibrosis during ductular reaction | https://www.ncbi.nlm.nih.gov/geo/query/acc.cgi?acc=GSE155981 | NCBI Gene Expression Omnibus, GSE155981 |

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
