## [Decision Letter]

**Acceptance summary:**

This study explores the roles of biliary epithelial cells and hepatocytes in repair of liver injury. They make the interesting observation that the loss of β-Catenin in both binary epithelial cells and hepatocytes results in persistent inflammation and fibrosis with persistent activation of NFκB, wheres the loss in just hepatocytes results in healing. This pathology was reminiscent of liver damage seen in cystic fibrosis leading the group to identify an interaction with CFTR. These studies will encourage others to explore the relative contributions of biliary epithelial cells and hepatocytes in liver injury and repair.

**Decision letter after peer review:**

Thank you for submitting your article "β-Catenin-NFκB-CFTR interactions in cholangiocytes regulate inflammation and fibrosis during ductular reaction" for consideration by *eLife*. Your article has been reviewed by 3 peer reviewers, and the evaluation has been overseen by Paul Noble as the Senior Editor and Reviewing Editor. The following individual involved in review of your submission has agreed to reveal their identity: Xin Chen (Reviewer #1).

Essential revisions:

1. The IHC images of β-Catenin in Figures 1C and 3C are very hard to interpret. A higher resolution images or inset images will be highly helpful.

2. Also, what happens to GS IHC in the two mouse Ctnnb1 KO model?

3. It would be helpful to quantify CD45(+) cells in Figures 5A and 5B.

4. For Figure7, does bCatenin and P65 interaction occur in cytoplasm or nuclear in BECs? Please clarify.

5. Sup. 1A showing that PanCK(+) cells are proliferating is very important. However, the staining is not entirely convincing (maybe due to the low resolution) that those PCNA(+) cells are also PanCK(+) (seem to be the adjacent cells are positive for PCNA). Perhaps better images or additional DR markers will be helpful.

6. In the introduction, the author mentioned that YAP has been implicated in DR. What happens to YAP in these Ctnnb1 KO liver samples? Some studies or discussion would be helpful.

7. Most of the experimental groups for KO mice analyzed only n=3 or 4, which caused a big variation and reduced statistical power. This point should be improved throughout the study.

8. Figure 2A,B examined fibrosis. Sirius red staining showed fibrosis is similar degrees in WT2 and KO2 mice 2 weeks after recovery and more fibrosis in KO2 3 months after recovery. The pictures are inconsistent with the quantification shown in Figure 2B.

9. Figure 2C examined the expression of fibrogenic genes, Col1a1 and TGFb2. However, these data did not support the statement in the text because of not statistically different. What are other collagen and TGF-b genes?

10. Figure 2D showed DR in WT2 and KO2 during the recovery. DR is also associated with liver regeneration. What is the hepatocyte proliferation here? IHC for DR should also be quantified.

11. Figure 2F showed Timp1 expression for understanding matrix degradation. What is MMP expression?

12. In Figure 4A,B, almost no fibrosis is observed in WT1, which is inconsistent with WT2 in Figure 2A. WT1 and WT2 should theoretically show similar degrees of fibrosis. Sirius red staining showed a significant reduction of fibrosis in KO1 after 2 weeks of recovery. However, the quantification (Figure 4B) did not show that. Also, Figure 4C collagen 2C data did not support the authors' concept.

13. In Figure 6, the authors assessed NF-κBp65 nuclear translocation mainly in CK19 positive cells. Also, in the text, the authors mentioned "NF-κB, the master regulator of immune cell response." Is NF-κB activation in CK19 positive cells related to immune cell infiltration that the study showed in Figure 5. If so, how? There is a gap between immune cell infiltration and NF-κB activation in BECs.

14. Figure 8 showed the relationship between b-catenin, NF-κBp65, and CFTR. This figure suggested CF livers had decreased b-catenin expression that enhances NF-κB activity, which induces the expression of NF-κB target genes. The authors suggested CF livers may have a similar phenotype caused in KO1 mice. Do KO1 livers have altered CFTR expression or activity? If the study suggests the interaction of b-catenin and NF-κB with CFTR, KO1 could have altered CFTR expression and some phenotypes related to CFTR inhibition.

15. The authors should also provide histologic examination of liver injury in mice by HandE staining.

16. To study the repair of liver damage in KO1 and KO2 mice, it is necessary to determine whether KO1 and KO2 mice will spontaneously cause liver damage. Therefore, it is important to provide groups of KO1 and KO2 mice without CDE diet.

17. There seems to be a significant fat accumulation in KO1 and KO2 mice compared to WTs mice on CDE diet for 2w in liver tissue section. Therefore, the author should also assess fat metabolism.

18. Line 176, "KO2" should be "KO1".

*Reviewer #1 (Recommendations for the authors):*

In this manuscript, Hu et al., analyzed DR in hepatocyte specific Ctnnb1 KO or BEC/hepatocyte specific Ctnnb1 KO mice. The studies identified a novel biochemical interaction between β-catenin, NF-κB and CTFR in BECs, which are implicated in DR induced inflammation and fibrosis.

The major strengths of the paper are the use of two distinct Ctnnb1 KO mouse models for comparison, with no obvious method/result weaknesses. The likely impact of the studies on liver regeneration is quite high. Overall this is an excellent study of interest to scientists who are interested in liver regeneration.*Reviewer #2 (Recommendations for the authors):*

Hu et al., investigated the role of b-catenin in BECs in the regulation of NF-κB activity and the recovery from CDE-induced chronic liver injury. Also, the study suggests b-catenin and NF-κB activation may have significant roles in CF livers. First, the study compared the recovery patterns from CDE-induced chronic liver injury in mice deficient in b-catenin in hepithelial cells and in hepatocytes only. The study found b-catenin activity is crucial for the recovery of chronic liver inflammation, fibrosis, and DR. The authors found reduced activity of b-catenin is associated with prolonged NF-κB activation in BECs. The authors also found that CF patient livers had reduced b-catenin expression and NF-κB activity, and that b-catenin, NF-κB, and CFTR form the complex and regulate NF-κB activity in BECs. Overall, the study should be of interest to the researchers who investigate liver regeneration, its relation to BEC biology, and cystic fibrosis.

Strengths:

1. The use of two different knockout mice; one is hepithelial cell-specific b-catenin knockout mice and the other is hepatocyte only-specific b-catenin knockout mice.

2. Understanding the recovery phase of chronic liver injury is important.

3. The crosstalk between b-catenin and NF-κB in BECs is of interest.

4. The interaction of b-catenin and NF-κB with CFTR is clinically relevant.

Weaknesses:

1. Most of the experimental groups for KO mice analyzed only n=3 or 4, which caused a big variation and reduced statistical power. This point should be improved throughout the study.

2. Figure 8 showed the relationship between b-catenin, NF-κBp65, and CFTR. This figure suggested CF livers had decreased b-catenin expression that enhances NF-κB activity, which induces the expression of NF-κB target genes. The authors suggested CF livers may have a similar phenotype caused in KO1 mice. However, it is unclear whether KO1 livers have altered CFTR expression or activity. If the study suggests the interaction of b-catenin and NF-κB with CFTR, KO1 could have altered CFTR expression and some phenotypes related to CFTR inhibition.

*Reviewer #3 (Recommendations for the authors):*

In this study, the authors studied hepatic injury and repair in two different genetic knockouts of β-catenin, one with β-catenin loss is hepatocytes and biliary epithelial cells (KO1), and another with loss in only hepatocytes (KO2) mice. These mice were challenged for 2w with choline-deficient, ethionine-supplemented (CDE) diet and allowed to recover on normal diet for 2w, 3m and 6m. KO2 show gradual liver repopulation with biliary epithelial cell-derived β-catenin-positive hepatocytes, and resolution of injury. KO1 showed persistent loss of β-catenin, NF-κB activation in biliary epithelial cells, progressive ductular reaction and fibrosis. They identified interactions of β-catenin, NF-κB and CFTR in biliary epithelial cells. Loss of CFTR or β-catenin led to NF-κB activation, ductular reaction and inflammation. Although data provided here by the authors could have interesting implications, there are several concerns about the research design.

Specific points:

1. The authors should also provide histologic examination of liver injury in mice by HandE staining.

2. To study the repair of liver damage in KO1 and KO2 mice, it is necessary to determine whether KO1 and KO2 mice will spontaneously cause liver damage. Therefore, it is important to provide groups of KO1 and KO2 mice without CDE diet.

3. There seems to be a significant fat accumulation in KO1 and KO2 mice compared to WTs mice on CDE diet for 2w in liver tissue section. Therefore, the authors should also assess fat metabolism.

---

## [Author Response]

Essential revisions:1. The IHC images of β-Catenin in Figures 1C and 3C are very hard to interpret. A higher resolution images or inset images will be highly helpful.

We have now updated figures 1C and 3C with better quality images for β-catenin immunohistochemistry. We have included 100x images here. We have also added higher magnification images (200x) as supplemental figure for both KO1 and KO2 models for β-catenin staining (Figure S1).

2. Also, what happens to GS IHC in the two mouse Ctnnb1 KO model?

That is an excellent point. We have added IHC for GS for both KO1 and KO2 livers at 3 and 6 months of recovery from 2-week CDE diet. We observed continued absence of GS in KO1 while GS progressively re-appears in zone-3 hepatocytes in KO2. This data is presented as a supplemental figure S2.

3. It would be helpful to quantify CD45(+) cells in Figures 5A and 5B.

This has now been included in Supplemental Figure 5 and shows statistically significantly higher inflammation in KO1 at 6 months as compared to all other time points although KO1 are always tending to have higher CD45 cells than WT1 at comparable time-points. This is in contrast to KO2 which progressively and significantly resolved CD45 infiltrate at 3 months onwards after recovery from CDE.

4. For Figure7, does bCatenin and P65 interaction occur in cytoplasm or nuclear in BECs? Please clarify.

Based on colocalization studies using immunofluorescence (now supplementary figure 9F), most of the interaction of β-catenin and p65 occurs in the cytoplasm of the BECs in vivo. in vitro data in SMCCc shows that most β-catenin and p65 are located in cytoplasmic compartment (Figure 7C). Upon β-catenin knockdown, there is a modest increase in nuclear p65 levels and activity, which is more pronounced following LPS treatment. This data also supports that β-catenin-p65 binding occurs in cytoplasm at baseline which prevents nuclear translocation of p65 spontaneously or after treatment with LPS. We have better clarified this in both results and discussion

5. Sup. 1A showing that PanCK(+) cells are proliferating is very important. However, the staining is not entirely convincing (maybe due to the low resolution) that those PCNA(+) cells are also PanCK(+) (seem to be the adjacent cells are positive for PCNA). Perhaps better images or additional DR markers will be helpful.

Supplementary Figure 1A (now Figure S4A) (right panel) shows PCNA^+^ cells to be present in ducts (PanCK^+^) as well as in nonductal (PanCK-) cells in the periportal area. The white arrowhead identifies only PanCK^+^; PCNA^+^ cells, which were carefully identified and quantified. We wanted to show a low magnification image to capture a larger representative area but we do quantify these images to present as a bar graph which is accurate.

6. In the introduction, the author mentioned that YAP has been implicated in DR. What happens to YAP in these Ctnnb1 KO liver samples? Some studies or discussion would be helpful.

YAP1 is seen mostly in biliary epithelial cells in an adult liver and contributes to their survival and proliferation. Its deletion from cholangiocytes disrupts homeostasis leading to bile duct paucity and cholestatic injury. Since there is increased DR in KO1 at 3 and 6 months of recovery from CDE, YAP1 levels are expected to increase simply due to their expression in these cells. Indeed we see increased YAP1 (Author response image 1) in DR in KO1. However, whether this increase in YAP1 in DR is contributing to the disease pathogenesis in our model requires additional studies such as generation of conditional KO of YAP1 along with β-catenin, which is out of scope of the current study.

**Author response image 1. sa2fig1:** 

7. Most of the experimental groups for KO mice analyzed only n=3 or 4, which caused a big variation and reduced statistical power. This point should be improved throughout the study.

We agree that our study uses n≥3 mice for control and KO groups and at various time points. This is chiefly due to allocation of the minimal numbers of mice to multiple time points for analysis that still allow us to perform statistics and allow some generalizable conclusions. During COVID-19 pandemic, there were additional restrictions placed on how many mice we were allowed to keep. Additionally, we believe that the variability seen in RT-PCR analysis is most likely due to diversity in cell composition of randomly selected liver pieces that could have varying degree of ductular reaction etc. We have been careful to not over-interpret results. We have been transparent about all significant and insignificant differences in various markers and mentioned trends versus significance, where applicable.

8. Figure 2A,B examined fibrosis. Sirius red staining showed fibrosis is similar degrees in WT2 and KO2 mice 2 weeks after recovery and more fibrosis in KO2 3 months after recovery. The pictures are inconsistent with the quantification shown in Figure 2B.

We apologize for this confusion. While we were careful to include the representative images, we have now replaced these images with better quality 100x images. We do observe a clear difference in Sirius Red staining (Red color) in KO2 as compared to WT2 at the time-points indicated as was assessed by densitometry for Sirius red staining. All statistics were done using one way ANOVA.

9. Figure 2C examined the expression of fibrogenic genes, Col1a1 and TGFb2. However, these data did not support the statement in the text because of not statistically different. What are other collagen and TGF-b genes?

Again, we are confident that the results are accurate and while the RT-PCR does not show significant difference in gene expression of these various genes indicative of ongoing fibrosis, the trends are clear. Eventually though, at 6 months, our key time point of concern, the fibrosis has subsided in KO2, and comparable to WT2. The possible reason for the variability in expression of genes within biological replicates is likely due to diversity in cell composition of randomly selected liver pieces for RT-PCR. One way ANOVA, which is more stringent than unpaired T test was used for significance.

10. Figure 2D showed DR in WT2 and KO2 during the recovery. DR is also associated with liver regeneration. What is the hepatocyte proliferation here? IHC for DR should also be quantified.

We have previously published hepatocyte proliferation in this model (PMID: 30215850). There is decrease in hepatocyte proliferation in KO2 following CDE injury and recovery due to lack of β-catenin, but as cholangiocyte-derived hepatocytes appear, they show more profound proliferation and gradually repopulate the KO2 liver. We have quantified IHC for panCK for DR for both figure 2D and 4D and data is presented as supplemental figure S3.

11. Figure 2F showed Timp1 expression for understanding matrix degradation. What is MMP expression?

While this is a good point, as the reviewers are aware, there are multiple MMPs and can be secreted my multiple cell types. Hence a comprehensive analysis of all MMPs might not be feasible. We assessed MMP9 and MMP13 on the available whole liver RNA and data is presented in (Author response image 2) . As can be appreciated, the expression of these two MMPs does not seem to be demonstrating any notable changes among the genotypes at the time points on samples that were available.

12. In Figure 4A,B, almost no fibrosis is observed in WT1, which is inconsistent with WT2 in Figure 2A. WT1 and WT2 should theoretically show similar degrees of fibrosis. Sirius red staining showed a significant reduction of fibrosis in KO1 after 2 weeks of recovery. However, the quantification (Figure 4B) did not show that. Also, Figure 4C collagen 2C data did not support the authors' concept.

We apologize for these errors and appreciate the reviewer’s astute comments. The images in figure 4A for WT1 have now been replaced by more representative images. The reviewer is also correct and WT1 and WT2 showed comparable but low fibrosis after CDE injury. The KO1 also continue to show higher fibrosis than controls at all time points and do not show any decrease in fibrosis at 2 weeks of recovery. A better image has been included and text has been corrected. The Sirius red quantification in figure 4B is indeed correct. However, the collagen data, like mentioned previously, shows higher trends of expression which did not achieved significance as assessed by one way ANOVA. This is likely due to variability of DR observed in pieces of tissue used for RNA isolation and RT-PCR.

13. In Figure 6, the authors assessed NF-κBp65 nuclear translocation mainly in CK19 positive cells. Also, in the text, the authors mentioned "NF-κB, the master regulator of immune cell response." Is NF-κB activation in CK19 positive cells related to immune cell infiltration that the study showed in Figure 5. If so, how? There is a gap between immune cell infiltration and NF-κB activation in BECs.

This is a valid point. We show using in vivo data first that absence of β-catenin in BECs led to increased nuclear p65 which was associated with increase expression of chemokines (Figure 6D and FigS8). However, since whole livers were used, we wanted to verify if this was also occurring in isolated BECs. For this we used SM-CC cells and upon β-catenin knockdown, showed increased p65 reporter activity along with increased nuclear p65. There was simultaneous increase in expression of similar chemokines like *CCl2* and *Cxcl5* that are known drivers of inflammation. Simultaneously we see increased CD45 positive cells in KO1 in periportal region where p65 is observed in reactive cholangiocytes. KO2 have no inflammation and no nuclear p65 in cholangiocytes. Taken together, NF-κB in cholangiocytes through expression of chemokines, is likely the driver of inflammation. We apologize for not making this clear but have now added clarity to this end in the Discussion section.

14. Figure 8 showed the relationship between b-catenin, NF-κBp65, and CFTR. This figure suggested CF livers had decreased b-catenin expression that enhances NF-κB activity, which induces the expression of NF-κB target genes. The authors suggested CF livers may have a similar phenotype caused in KO1 mice. Do KO1 livers have altered CFTR expression or activity? If the study suggests the interaction of b-catenin and NF-κB with CFTR, KO1 could have altered CFTR expression and some phenotypes related to CFTR inhibition.

This is an important point. What we are proposing is that loss of function mutations in CFTR leads to its inability to bind β-catenin in BECs which leads to its destabilization. The ensuing decrease in β-catenin protein allows for p65 to get activated more readily in the presence of cytokines to lead to NF-κB activation and downstream signaling. We do not believe that β-catenin loss impacts CFTR levels although we are unsure if that impacts CFTR function which needs to be investigated in the future. Since CFTR is expressed in BECs only, we used whole tissue lysate from WT1 and KO1 at baseline and at 6m recovery to test any changes in total CFTR levels. We did not find any changes in total levels of CFTR protein. This data has been included as supplemental figure 10 and text added in Results section and in the Discussion section as well.

15. The authors should also provide histologic examination of liver injury in mice by HandE staining.

We are including a histology image, Author response image 3, to the reviewers. We don’t think it adds to the overall message and clarity to the study beyond the already included immunohistochemistry and special stains. We would be happy to include it as an additional supplemental figure (10 supplemental figures thus far), if advised.

**Author response image 3. sa2fig3:** 

16. To study the repair of liver damage in KO1 and KO2 mice, it is necessary to determine whether KO1 and KO2 mice will spontaneously cause liver damage. Therefore, it is important to provide groups of KO1 and KO2 mice without CDE diet.

WT2 and KO2 fed normal diet and without any CDE injury were assessed for serum biochemistry. Insignificant changes between these two genotypes in serum levels of ALT were observed. Intriguingly, there were a subset of KO2 mice maintained on normal diet and without any previous CDE diet induced injury, that showed higher levels of total bilirubin. These changes were insignificant as assessed by one way ANOVA. It is important to mention that KO2 fed CDE diet for 2 weeks and followed on normal diet for various times, begin to show repopulation of these livers by BEC-derived β-catenin-positive hepatocytes such that KO2 are no longer KOs. The serum biochemistry of these recovered mice is unremarkable and indistinguishable from WT2 and hence the comparison between KO2 on normal diet versus 2w CDE and normal diet recovery might not be necessary.

KO1 and WT1 models have been published by us previously. We showed that there is no spontaneous injury but with increasing age (8 months), there is mild increase in serum BR (Tan X, Behari J, Cieply B, Michalopoulos GK, Monga SP. Conditional deletion of β-catenin reveals its role in liver growth and regeneration. *Gastroenterology* 131, 1561-1572 (2006)). In the current study, we examined the serum biochemistry from KO1 and WT1 mice on normal diet. We found insignificant differences in serum ALT and total bilirubin between WT1 and KO1 maintained on normal diet.

Both serum ALT and BR data from normal diet fed WT2, KO2, WT1 and KO1 are included as Author response image 4 but not in the actual manuscript. If editors and reviewers prefer that we include these data in the article as a supplemental figure, we will be happy to comply.

**Author response image 4. sa2fig4:** 

17. There seems to be a significant fat accumulation in KO1 and KO2 mice compared to WTs mice on CDE diet for 2w in liver tissue section. Therefore, the author should also assess fat metabolism.

Choline deficiency is well known to cause steatosis and is known to be a contributing mechanism of steatosis even in patients of NAFLD. CDE diet causes varying degrees of steatohepatitis and injury in both WT and KO, although it is greater in KO. We have reported this in the first study (PMID:30215850). But addressing pathogenesis of steatosis is out of scope of the current study, which is mainly focused on examining the molecular basis of reactive ductules that leads to inflammation and fibrosis.

18. Line 176, "KO2" should be "KO1".

We apologize for this error and are grateful for pointing it out. This has been corrected.